# Measuring Interpretability of Neural Policies of Robots with Disentangled Representation

**Tsun-Hsuan Wang, Wei Xiao, Tim Seyde, Ramin Hasani, Daniela Rus**
Massachusetts Institute of Technology (MIT)

**Abstract:** The advancement of robots, particularly those functioning in complex human-centric environments, relies on control solutions that are driven by machine learning. Understanding how learning-based controllers make decisions is crucial since robots are often safety-critical systems. This urges a formal and quantitative understanding of the explanatory factors in the interpretability of robot learning. In this paper, we aim to study interpretability of compact neural policies through the lens of disentangled representation. We leverage decision trees to obtain *factors of variation* [1] for disentanglement in robot learning; these encapsulate skills, behaviors, or strategies toward solving tasks. To assess how well networks uncover the underlying task dynamics, we introduce interpretability metrics that measure disentanglement of learned neural dynamics from a concentration of decisions, mutual information and modularity perspective. We showcase the effectiveness of the connection between interpretability and disentanglement consistently across extensive experimental analysis.

**Keywords:** Interpretability, Disentangled Representation, Neural Policy

## 1 Introduction

Interpretability of learning-based robot control is important for safety-critical applications as it affords human comprehension of how the system processes inputs and decides actions. In general, achieving interpretability is difficult for learning-based robot control. The robot learning models make decisions without being explicitly programmed to perform the task and are often very large, thus it

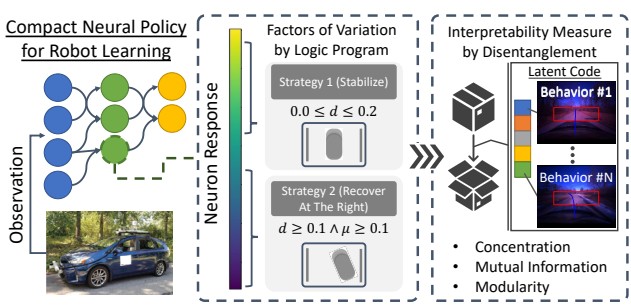

Figure 1: Understand robot behaviors by extracting logic programs as factors of variation to measure interpretability with disentanglement.

is impossible to synthesize and explain their reasoning processes. This lack of transparency, often referred to as the "black box" problem, makes it hard to interpret the workings of learning-based robot control systems. Understanding why a particular decision was made or predicting how the system will behave in future scenarios remains a challenge, yet critical for physical deployments.

Through the lens of representation learning, we assume that neural networks capture a set of processes that exist in the data distribution; for robots, they manifest learned skills, behaviors, or strategies, which are critical to understand the decision-making of a policy. However, while these *factors of variation* [1] (e.g., color or shape representations) are actively studied in unsupervised learning for disentangled representation, in robot learning, they are less well-defined and pose unique challenges due to the intertwined correspondence of neural activities with emergent behaviors unknown a priori. In the present study, we aim to (i) provide a useful definition of factors of variation for policy learning, and (ii) explore how to uncover dynamics and factors of variation quantitatively as a measure of interpretability in compact neural networks for closed-loop end-to-end control applica-

7th Conference on Robot Learning (CoRL 2023), Atlanta, USA.

tions. In this space, an entanglement corresponding to multiple neurons responsible for an emergent behavior can obstruct the interpretation of neuron response even with a small number of neurons [2, 3, 4, 5]. To this end, the disentanglement of learned representations [6, 7, 8] in compact neural networks is essential for deriving explanations and interpretations for neural policies.

We posit that each neuron should learn an abstraction (factor of variation) related to a specific strategy required for solving a sub-component of a task. For example, in locomotion, one neuron may capture periodic gait, where the numerical value of the neuron response may be aligned with different phases of the gait cycle; another neuron may account for recovery from slipping. Retrieving the abstraction learned by a neuron is, however, non-trivial. Directly observing the neuron response along with sensory information provided as input to the policy can be extremely inefficient and tedious for identifying behaviors and interpreting decision-making.

In this work, our objective is to formulate an abstraction that represents the decision-making of a parametric policy to quantify the interpretability of learned behaviors, specifically from the perspective of disentangled representations. To this end, we make the following key contributions:

- Provide a practical definition of factor of variation for robot learning by programmatically extracting decision trees from neural policies, in the form of logic program grounded by world states.
- Introduce a novel set of quantitative metrics of interpretability to assess how well policies uncover task structures and their factors of variation by measuring the disentanglement of learned neural dynamics from a concentration of decisions, mutual information, and modularity perspective.
- Experiment in a series of end-to-end policy learning tasks that (a) showcase the effectiveness of leveraging disentanglement to measure interpretability, (b) demonstrate policy behaviors extracted from neural responses, (c) unveil interpretable models through the lens of disentanglement.

## 2 Related Work

**Compact neural networks.** Compact neural networks are ideal in resource constraints situations such as robotics and by nature easier to interpret due to a smaller number of neurons [2, 4]. Compact networks can be obtained by pruning [9] or compressing neural networks in end-to-end training [10]. Regularization has also been used to generate compact neural networks [11]. Compact representation of features can be learned using discriminative masking [12]. Neural Ordinary Differential Equations have also been used for learning compact network [13, 4]. In this work, we formally study the interpretability of compact neural policies through the lens of disentangled representation.

**Interpretable neural networks.** An interpretable neural network could be constructed from a physically comprehensible perspective [14, 15]. Knowledge representation is used to obtain interpretable Convolutional Neural Network [16]. An active line of research focuses on dissecting and analyzing trained neural networks in a generic yet post-hoc manner [17, 18, 19, 20]. Another active line of research is to study disentangled explanatory factors in learned representation [8]. A better representation should contain information in a compact and interpretable structure [1, 21]. Unlike prior works that study disentanglement based on *factors of variations* such as object types, there is no notion of ground-truth factors in robot learning and thus we propose to use decision trees to construct pseudo-ground-truth factors that capture emergent behaviors of robot for interpretability analysis.

**Interpretability in policy learning.** Explainable AI has been recently extended to policy learning like reinforcement learning [22] or for human-AI shared control settings [23]. One line of research analyzes multi-step trajectories from the perspective of options or compositional skills [24, 25, 26, 27, 28, 29]. A more fine-grained single-step alternative is to extract policies via imitation learning to interpretable models like decision tree [30]. Another line of work directly embeds the decision tree framework into the learning-based model to strike a balance between expressiveness and interpretability [31, 32, 33]. Explanation of policy behaviors can also be obtained by searching for abstract state with value function [34] or feature importance [35]. In this work, we aim to offer a new perspective of disentangled representation to measure interpretability in robot policy learning.

## 3 Method

---

**Algorithm 1** Extract Abstraction via Decision Tree

---

**Data** Trajectories rollout from a compact neural policy $\mathcal{D}_{dt} = \{(o_0, s_0, a_0, z_0, \dots)_j\}_{j=1}^N$
**Result** Interpreters of neuron response $\{f_\mathcal{S}^i\}_{i \in \mathcal{I}}$
**for** $i \in \mathcal{I}$ **do**
    Train a decision tree $T_{\theta^i}$ from state $\{s_t\}$ to neural response $\{z_t^i\}$.
    Collect dataset $\mathcal{D}_{dp}$ with neuron response $\{z_t^i\}$ and decision paths $\{\mathcal{P}_{s_t}^i\}$.
    Train neuron response classifier $q_{\phi^i} : \mathbb{R} \to \{\mathcal{P}\}$ with $\mathcal{D}_{dp}$.
    Obtain decision path parser $r^i : \{\mathcal{P}\} \to \mathcal{L}$ by tracing out $\{\mathcal{P}_k^i\}_{k=1}^{K^i}$ in $T_{\theta^i}$.
    Construct the mapping $f_\mathcal{S}^i = r^i \circ q_{\phi^i}$.
**end for**

---

In this section, we describe how to obtain factor of variation by predicting logic programs from neuron responses that reflect the learned behavior of the policies (Section 3.1), followed by a set of quantitative measures of interpretability in the lens of disentanglement (Section 3.2).

## 3.1 Extracting Abstraction via Decision Tree

Our goal is to formulate a logic program that represents the decision-making of a parametric policy to serve as an abstraction of learned behaviors, summarized in Algorithm 1. First, we describe a decision process as a tuple $\{\mathcal{O}, \mathcal{S}, \mathcal{A}, P_a, h\}$, where at a time instance $t$, $o_t \in \mathcal{O}$ is the observation, $s_t \in \mathcal{S}$ is the state, $a_t \in \mathcal{A}$ is the action, $P_a : \mathcal{S} \times \mathcal{A} \times \mathcal{S} \to [0, 1]$ is the (Markovian) transition probability from current state $s_t$ to next state $s_{t+1}$ under action $a_t$, and $h : \mathcal{S} \to \mathcal{O}$ is the observation model. We define a neural policy as $\pi : \mathcal{O} \to \mathcal{A}$ and the response of neuron $i \in \mathcal{I}$ as $\{z_t^i \in \mathbb{R}\}_{i \in \mathcal{I}}$, where $\mathcal{I}$ refers to a set of neurons to be interpreted. For each neuron $i$, we aim to construct a mapping that infers a logic program from neuron response, $f_\mathcal{S}^i : \mathbb{R} \to \mathcal{L}$, where $\mathcal{L}$ is a set of logic programs grounded on environment states $\mathcal{S}$. Note that $f_\mathcal{S}^i$ does not take the state as an input as underlying states may be inaccessible during robot deployment. In the following discussion, we heavily use the notation $\mathbf{P}_*^i$ for the decision path associated with the $i$'th neuron, where the subscript $*$ refers to the dependency on state if with parenthesis (like $_{(s_t)}$) and otherwise indexing based on the context.

**From states to neuron responses.** Decision trees are non-parametric supervised learning algorithms for classification and regression. Throughout training, they develop a set of decision rules based on thresholding one or a subset of input dimensions. The relation across rules is described by a tree structure with the root node as the starting point of the decision-making process and the leaf nodes as the predictions. The property of decision trees to convert data for decision making to a set of propositions is a natural fit for state-grounded logic programs. Given a trained neural policy $\pi$, we collect a set of rollout trajectories $\mathcal{D}_{\text{dt}} = \{\tau_j\}_{j=1}^N$, where $\tau_j = (o_0, s_0, a_0, z_0, o_1, \dots)$. We first train a decision tree $T_{\theta^i}$ to predict the $i$th neuron response from states,

$$\theta^{i*} = \arg\min_{\theta^i} \sum_{(s_t, z_t^i) \in \mathcal{D}_{\text{dt}}} L_{\text{dt}}(\hat{z}_t^i, z_t^i), \quad \text{where } \hat{z}_t^i = T_{\theta^i}(s_t) \tag{1}$$

where $L_{\text{dt}}$ represents the underlying classification or regression criteria. The decision tree $T_{\theta^i}$ describes relations between the neuron responses and the relevant states as logical expressions. During inference, starting from the root node, relevant state dimensions will be checked by the decision rule in the current node and directed to the relevant lower layer, finally arriving at one of the leaf nodes and providing information to regress the neuron response. Each inference traces out a route from the root node to a leaf node. This route is called a decision path. A decision path consists of a sequence of decision rules defined by nodes visited by the path, which combine to form a logic program,

$$\bigwedge_{n \in \mathcal{P}_{(s_t)}^i, j=g(n)} (s_t^j \leq c_n) \longleftrightarrow \text{Behavior extracted from } \hat{z}_t^i \text{ via } T_{\theta^i} \tag{2}$$

where $\wedge$ is the logical AND, $\mathcal{P}_{(s_t)}^i$ is the decision path of the tree $T_{\theta^i}$ that takes $s_t$ as inputs, $g$ gives the state dimension used in the decision rule of node $n$ (assume each node uses one feature for notation simplicity), and $c_n$ is the threshold at node $n$.

**From neuron responses to decision paths.** So far, we recover a correspondence between the neuron response $z_t$ and the state-grounded program based on decision paths $\mathcal{P}_{(s_t)}^i$; however, this is not

sufficient for deployment since the decision tree $T_{\theta^i}$ requires as input the ground-truth state and not the observable data to the policy (like $o_t, z_t$). To address this, we find an inverse of $T_{\theta^i}$ with neuron responses as inputs and pre-extracted decision paths as classification targets. Based on the inference process of $T_{\theta^i}$, we can calculate the numerical range of neuron responses associated with a certain decision path $\mathcal{P}^i_{(s_t)}$ from the predicted $\hat{z}_t$ and then construct the pairs of $z_t$ and $\mathcal{P}^i_{s_t}$. We collect another dataset $\mathcal{D}_{\mathrm{dp}}$ and train a classifier $q_{\phi^i}$ to predict decision paths from neuron responses,

$$\phi^{i*} = \arg\min_{\phi^i} \sum_{(z^i_t, \mathcal{P}^i_{(s_t)}) \in \mathcal{D}_{\mathrm{dp}}} L_{\mathrm{dp}}(q_{\phi^i}(z^i_t), \mathcal{P}^i_{(s_t)}) \tag{3}$$

where $L_{\mathrm{dp}}$ is a classification criterion. While $\mathcal{P}^i_{(s_t)}$ is state-dependent, there exists a finite set of decision paths $\{\mathcal{P}^i_k\}^{K^i}_{k=1}$ given the generating decision tree. We define the mapping from the decision tree to the logic program as $r : \{\mathcal{P}\} \to \mathcal{L}$, which can be obtained by tracing out the path as described above. Overall, the desired mapping is readily constructed as $f^i_{\mathcal{S}} = r^i \circ q_{\phi^i}$.

## 3.2 Quantitative Measures of Interpretability

Programmatically extracting decision trees for constructing a mapping from the neuron response to a logic program offers a representation that facilitates the interpretability of compact neural policies. Furthermore, building on the computational aspect of our approach, we can quantify the interpretability of a policy with respect to several metrics through the lens of disentanglement.

**A. Neuron-Response Variance.** Given decision paths $\{\mathcal{P}^i_k\}^{K^i}_{k=1}$ associated with a tree $T_{\theta^i}$ at the $i$th neuron, we compute the normalized variance of the neuron response averaged across decision paths,

$$\frac{1}{|\mathcal{I}|} \sum_{i \in \mathcal{I}} \frac{1}{K^i} \sum^{K^i}_{k=1} \mathrm{Var}_{\substack{(s_t, z^i_t) \in \mathcal{D}_{\mathrm{dt}} \\ t \in \{u | \mathcal{P}^i_{(s_u)} = \mathcal{P}^i_k\}}} \left[ \frac{z^i_t}{Z^i} \right] \tag{4}$$

where $Z^i$ is a normalization factor that depends on the range of response of the $i$th neuron. The set $\{u | \mathcal{P}^i_{(s_u)} = \mathcal{P}^i_k\}$ contains all time steps that exhibit the same behavior as entailed by $\mathcal{P}^i_k$. For example, suppose we have a trajectory consisting of behaviors including walking and running, and that walking is depicted as $\mathcal{P}^i_k$, the set refers to all time steps of walking. This metric captures the concentration of the neuron response that corresponds to the same strategy represented by the logic program defined by $T_{\theta^i}$. In practice, we discretize all neuron responses to $N$ bins, compute the index of bins to which a value belongs, divide the index by $N$ and compute their variance.

**B. Mutual Information Gap.** Inspired by [21, 8], we integrate the notion of mutual information in our framework to extend disentanglement measures for unsupervised learning to policy learning. Specifically, while previous literature assumes known ground-truth factors for disentanglement such as object types, viewing angles, etc., there is no straightforward equivalence in neural policies since the emergent behaviors or strategies are unknown a priori. To this end, we propose to leverage the decision path sets to construct pseudo-ground-truth factors $\mathcal{M}_{dp} = \bigcup_{i \in \mathcal{I}} \{\mathcal{P}^i_k\}^{K^i}_{k=1} = \{\mathcal{P}_k\}^K_{k=1}$. Note that there may be a correlation across decision paths, i.e., $P(\mathcal{P}_i, \mathcal{P}_j) \neq P(\mathcal{P}_i)P(\mathcal{P}_j)$ for $i \neq j$. For example, one decision path corresponding to a logic program of the robot moving forward at high speed has a correlation to another decision path for moving forward at low speed. This may occur because a neuron of a policy can learn arbitrary behaviors. However, this leads to a non-orthogonal ground-truth factor set and can be undesirable since high correlations of a neuron to multiple ground-truth factors (e.g., $I[z^i; \mathcal{P}_i]$ and $I[z^i; \mathcal{P}_j]$ are large) can result from not only entanglement of the neuron but also the correlation between factors (e.g., $I[\mathcal{P}_i; \mathcal{P}_j]$ is large). Hence, this urges the need to calibrate mutual information for computing disentanglement measures. We start by adapting the Mutual Information Gap (MIG) [21] to our framework:

$$\frac{1}{K} \sum^K_{k=1} \frac{1}{H[\mathcal{P}_k]} \left( I[z^{i*}; \mathcal{P}_k] - \max_{j \neq i^*} I[z^j; \mathcal{P}_k] - I[z^j; \mathcal{P}_k; \mathcal{P}_{k^j}] \right) \tag{5}$$

where $H$ is entropy, $I$ is interaction information that can take an arbitrary number of variables (with 2 being mutual information), $i^* = \arg\max_i I[z^i; \mathcal{P}_k]$, and $k^j = \arg\max_l I[z^j; \mathcal{P}_l]$. Intuitively,

Table 1: Quantitative results of classical control.

| Network Architecture | Disentanglement | | | Explanation Size ↓ | | Cognitive Chunks ↓ |
| --- | --- | --- | --- | --- | --- | --- |
| | Variance ↓ | MI-Gap ↑ | Modularity ↑ | Vertical | Horizontal | |
| FCs | $0.0242^{.005}$ | $0.3008^{.025}$ | $0.9412^{.014}$ | $5.00^{.46}$ | $1.91^{.14}$ | $1.65^{.28}$ |
| GRU | $0.0329^{.004}$ | $0.2764^{.062}$ | $0.9096^{.022}$ | $4.90^{.80}$ | $1.96^{.17}$ | $1.65^{.25}$ |
| LSTM | $\mathbf{0.0216}^{.003}$ | $0.2303^{.024}$ | $0.9355^{.008}$ | $4.75^{.39}$ | $2.02^{.12}$ | $1.90^{.14}$ |
| ODE-RNN | $0.0287^{.007}$ | $0.3062^{.041}$ | $0.9376^{.017}$ | $4.90^{.38}$ | $1.93^{.15}$ | $1.80^{.27}$ |
| CfC | $0.0272^{.004}$ | $0.2892^{.111}$ | $0.9067^{.039}$ | $4.70^{.65}$ | $1.82^{.33}$ | $1.50^{.47}$ |
| NCP | $0.0240^{.008}$ | $\mathbf{0.3653}^{.052}$ | $\mathbf{0.9551}^{.019}$ | $\mathbf{3.45}^{.83}$ | $\mathbf{1.51}^{.33}$ | $\mathbf{1.30}^{.32}$ |

Table 2: Alignment between disentanglement and explanation quality in classical control.

| Re-signed Rank Correlation ↑ | Explanation Size | | Cognitive Chunks |
| --- | --- | --- | --- |
| | Vertical | Horizontal | |
| Variance | -0.146 | 0.002 | 0.040 |
| MI-Gap | **0.427** | **0.505** | **0.449** |
| Modularity | -0.114 | 0.156 | 0.032 |

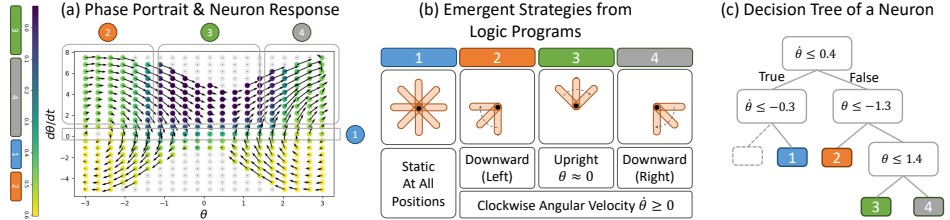

Figure 2: In classical control (Pendulum): (a) Phase portrait with empirically measured closed-loop dynamics and neuron response. Each arrow and colored dot are the results averaged around the binned state space. (b) Emergent strategies from logic programs. (c) Decision tree extracted for command neuron 3 in NCP.

this measures the normalized difference between the highest and the second-highest mutual information of each decision path with individual neuron activation, i.e., how discriminative the correlation between the neuron response is with one decision path as opposed to the others. For example, neuron response correlated to multiple factors of variation will have lower MIG than those to one only. The last term $I[z^j; \mathcal{P}_k; \mathcal{P}_{k^j}]$ is for calibration and captures the inherent correlation between $z^j$ and $\mathcal{P}_k$ resulted from potentially nonzero $I[\mathcal{P}_k; \mathcal{P}_{k^j}]$ with $\mathcal{P}_{k^j}$ being a proxy random variable of $z^j$ in the ground-truth factor set. We show how to compute $I[z^j; \mathcal{P}_k] - I[z^j; \mathcal{P}_k; \mathcal{P}_{k^j}]$ in Appendix Section C.

**C. Modularity.** We compute modularity scores from [36] with the same calibration term,

$$\frac{1}{\mathcal{I}} \sum_{i \in \mathcal{I}} 1 - \frac{\sum_{k \neq k^*} (I[z^i; \mathcal{P}_k] - I[z^i; \mathcal{P}_k; \mathcal{P}_{k^*}])^2}{(K-1)I[z^i; \mathcal{P}_{k^*}]^2}, \tag{6}$$

where $k^* = \arg\max_l I[z^i; \mathcal{P}_l]$. For a ideally modular representation, each neuron will have high mutual information to a single factor of variation and low mutual information with all the others. Suppose for each neuron $i$ has the best "match" with a decision path (ground-truth factor) $k^*$, non-modularity of that neuron is computed as the normalized variance of mutual information between its neuron response and all non-matched decision paths $\{\mathcal{P}_k\}_{k \neq k^*}$. In practice, we discretize neuron responses into $N$ bins to compute discrete mutual information.

# 4 Experiments

We conduct a series of experiments in various policy-learning tasks to answer the following: (i) *How effective is disentanglement to measure the interpretability of policies?* (ii) *What can we extract from neural responses?* (iii) *What architecture is more interpretable through the lens of disentanglement?*

## 4.1 Setup

**Network architecture.** We construct compact neural networks for each end-to-end learning to control task. For all tasks, our networks are constructed by the following priors: (i) Each baseline network is supplied with a perception backbone (e.g., a convolutional neural network) (ii) We construct policies based on different compact architectures that take in feature vectors from the perception backbone and output control with comparable cell counts (instead of actual network size in memory as we assess interpretability metrics down to cell-level). The perception backbone is followed by a neural controller designed by compact feed-forward and recurrent network architectures including fully-connected network (**FCs**), gated recurrent units (**GRU**) [37], and long-short term memory (**LSTM**) [38]. Additionally, we include advanced continuous-time baselines designed by ordinary differential equations such as **ODE-RNN** [39], closed-form continuous-time neural models (**CfCs**) [40], and neural circuit policies (**NCPs**) [4]. We interpret the dynamics of the neurons in the last layer before the output in FCs, the command-neuron layer of NCPs, and the recurrent state of the rest. We then extract logic programs and measure interpretability with the proposed metrics.

Table 3: Quantitative results of locomotion.

| Network Architecture | Disentanglement | | | Explanation Size ↓ | | Cognitive Chunks ↓ |
|---|---|---|---|---|---|---|
| | Variance ↓ | MI-Gap ↑ | Modularity ↑ | Vertical | Horizontal | |
| FCs | $0.0187^{.002}$ | $0.1823^{.013}$ | $0.9622^{.008}$ | $5.66^{.46}$ | $2.54^{.12}$ | $4.02^{.55}$ |
| GRU | $0.0259^{.002}$ | $0.1830^{.022}$ | $0.9713^{.009}$ | $5.78^{.39}$ | $2.52^{.08}$ | $3.94^{.35}$ |
| LSTM | $0.0108^{.002}$ | $0.1453^{.025}$ | $0.9600^{.002}$ | $5.62^{.31}$ | $2.52^{.10}$ | $3.92^{.23}$ |
| ODE-RNN | $0.0210^{.003}$ | $0.1880^{.029}$ | $0.9701^{.007}$ | $6.00^{.50}$ | $2.57^{.11}$ | $4.16^{.43}$ |
| CfC | $0.0234^{.004}$ | $0.1596^{.019}$ | $0.9628^{.009}$ | $5.94^{.15}$ | $2.58^{.04}$ | $4.20^{.32}$ |
| NCP | $\mathbf{0.0107^{.001}}$ | $\mathbf{0.2164^{.042}}$ | $\mathbf{0.9791^{.005}}$ | $\mathbf{3.94^{.25}}$ | $\mathbf{2.08^{.02}}$ | $\mathbf{2.72^{.18}}$ |

Table 4: Alignment between disentanglement and explanation quality in locomotion.

| Re-signed Rank Correlation ↑ | Explanation Size | | Cognitive Chunks |
|---|---|---|---|
| | Vertical | Horizontal | |
| Variance | **0.512** | 0.456 | 0.443 |
| MI-Gap | 0.422 | **0.504** | **0.481** |
| Modularity | 0.170 | 0.180 | 0.173 |

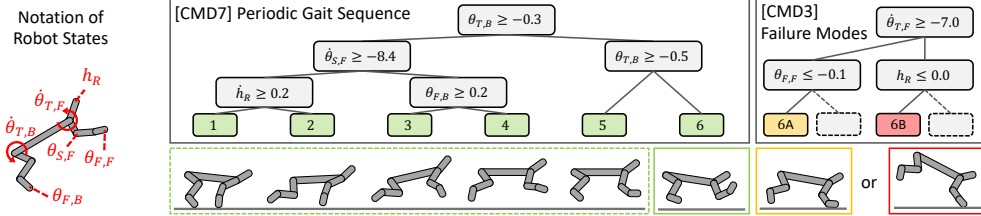

Figure 3: Neural activations along a gait sequence on HalfCheetah [43]. We focus on neurons 7 and 3 for illustration. Neuron 7 exhibits a periodic activation pattern that reacts to different phases of the gait cycle (left). Neuron 3 displays peak activity in situations with the potential to destabilize gait, such as early touchdown (left) and forward flipping (right). Our approach aids in failure detection by monitoring key neurons' responses.

**Evaluation.** To evaluate the effectiveness of measuring interpretability through the lens of disentanglement, we adopt the metrics proposed in [41], which studies human interpretability of decision sets [42] (a representation of explanation similar to that in this work). They show human response time and subjective satisfaction are highly correlated with *explanation size* and *cognitive chunks*. Explanation size consists of vertical size , the number of cases (the number of decision paths per neuron), and horizontal size, the complexity of each case (the length of each decision path). Cognitive chunks refer to the presentation of newly defined concepts, which we quantify as the introduction of new symbols in the logic program. Furthermore, we measure the alignment of disentanglement quantification and the above-mentioned explanation quality metrics. We compute *re-signed rank correlation* by re-signing Spearman's rank correlation coefficient to make larger values always refer to better alignment, e.g., given higher modularity being better while lower explanation size being better, better alignment corresponds to negative correlation and we thus negate the coefficient.

### 4.2 Classical Control

**Environment and policy learning.** We use the OpenAI Gym Classical Control Pendulum task [43]. The environment has simple yet nonlinear dynamics and allows for straightforward visualization of the entire state space. The environment states include $\theta$ (joint angle) and $\dot{\theta}$ (joint angular velocity). $\theta$ is in the range of $\pm\pi$ with $\theta = 0$ as the upright position. $\dot{\theta}$ is along the clockwise direction. The control is $u$ (joint torque). The goal is to stabilize at the upright position ($\theta = \dot{\theta} = 0$) with limited control energy consumption ($u \downarrow$). We use Proximal Policy Optimization (PPO) [44] to train the policy with early stop by reaching episode reward -500 or a maximal number of training iterations. We run each model with 5 different random seeds and report average results.

**Quantitative analysis.** Table 1 shows that, among all models, NCP achieves the best performance in disentanglement and explanation quality (i.e., explanation size and cognitive chunks), suggesting that it is more interpretable from the perspective of both our work and [41]. Beyond alignment of the best performance, Table 2 indicates the consistency of overall ranking between disentanglement and explanation quality. We found that while variance and modularity are (partially) aligned in the best performance in Table 1, only the mutual information gap is correlated to explanation quality in the overall ranking. Another interesting finding is that CfCs have the lowest logic conflict. By empirically checking the decision trees, they construct non-trivial but highly-overlapping decision paths, thus leading to considerably fewer conflicts in logic programs across neurons.

**Neuron responses and underlying behaviors.** While all models learn reasonable strategies, as exemplified by focusing on the sign of $\theta$ and $\dot{\theta}$, we now dive deeper into understanding individual neural dynamics. To this end, we focus on NCPs as they provide a lower variance from the disentanglement perspective in their logic programs. We found different neurons roughly subdivide the state space into quadrants and focus on their respective subsets. In Figure 2, we show the interpretability

Table 5: Quantitative results of visual servoing.

| Network Architecture | Disentanglement | | | Explanation Size ↓ | | Cognitive Chunks ↓ |
|---|---|---|---|---|---|---|
| | Variance ↓ | MI-Gap ↑ | Modularity ↑ | Vertical | Horizontal | |
| FCs | 0.0124 | 0.1354 | 0.9704 | 4.88 | 2.36 | 2.88 |
| GRU | 0.0158 | 0.1614 | 0.9801 | 3.88 | **1.94** | **1.75** |
| LSTM | 0.0172 | 0.1950 | **0.9851** | 4.25 | 2.06 | 2.38 |
| ODE-RNN | 0.0151 | 0.1588 | 0.9766 | 5.25 | 2.24 | 2.75 |
| CfC | 0.0191 | 0.1391 | 0.9677 | 5.50 | 2.43 | 3.00 |
| NCP | **0.0068** | **0.3902** | 0.9770 | **4.12** | 2.00 | 1.88 |

Table 6: Alignment between disentanglement and explanation quality in visual servoing.

| Re-signed Rank Correlation ↑ | Explanation Size | | Cognitive Chunks |
|---|---|---|---|
| | Vertical | Horizontal | |
| Variance | 0.371 | 0.314 | 0.314 |
| MI-Gap | **0.657** | **0.771** | **0.771** |
| Modularity | 0.257 | 0.314 | 0.314 |

(a) Interpretation of a single neuron

(b) Front-view image (observation) at different neuron activation

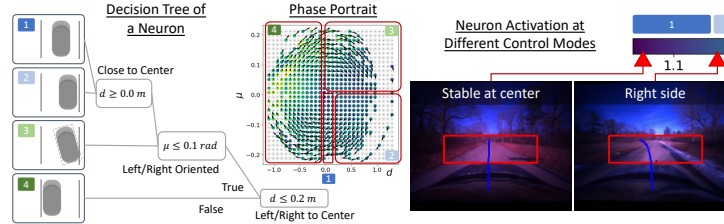

Figure 4: Explanation of neural policies for end-to-end visual servoing (Image-based Driving). (a) Phase portrait of local heading error $\mu$ and lateral deviation $d$ with empirically measured mean neuron response and closed-loop dynamics. (b) Front-view images retrieved based on neuron response.

analysis of command neuron 3 as an example. This neuron developed fine-grained strategies for different situations like swinging clockwise in the right or left downward positions, and stabilizing positive angular velocity around the upright position, as shown in Figure 2 (b)(c). We further provide phase portrait in Figure 2 (a). The arrows indicate empirically measured closed-loop dynamics (with control from the policy) and the color coding indicates average neuron response at a specific state from evaluation. The color of the neuron response (corresponding to logic programs) and the arrows (which implicitly capture actions) highlight different fine-grained strategies in the phase portrait. Notably, this finding applies not just to NCPs but also to other networks with similar functions.

### 4.3 Locomotion

**Environment and policy learning.** We consider a planar locomotion task based on OpenAI Gym's HalfCheetah environment [43]. The agent is rewarded for forward locomotion based on a simple base velocity reward. We optimize our policies with PPO until a maximum number of episodes has been reached. For each model, we run five trials with different random seeds and report average results. Here, our objective is to extend our interpretability framework to a higher-dimensional control task. Specifically, we investigate whether our approach is capable of extracting consistent single-neuron activation patterns that align with individual phases of a periodic gait cycle.

**Quantitative analysis.** In Table 3, we observe consistent results with classical control that NCP achieves the best performance in disentanglement and explanation quality. We further observe that the LSTM achieves a desirable low disentanglement variance comparable to NCPs. LSTMs and most networks compared to NCPs on the other hand show a lower Mutual Information Gap. This suggests that in these networks neuron responses are concentrated for different decision paths but not quite identifiable from a probabilistic perspective, as certain neuron activation cannot be uniquely mapped to a decision path. Besides, Table 4 shows consistent findings that the mutual information gap has the best ranking correlation with explanation quality. As opposed to classical control, the ranking correlation is overall much higher and we hypothesize that more complex tasks may yield better alignment in overall ranking between disentanglement and explanation quality.

**Exploring Gait Pattern.** As the state space is significantly larger than the Pendulum task, we complement our quantitative interpretability results with qualitative results that focus on two exemplary neurons, namely command neurons (CMDs) 3 and 7. Figure 3 provides the extracted decision trees for CMDs 7 (left) and 3 (right). We find that the former displays periodic activation patterns that align very well with individual phases of regular gait. In particular, it leverages position readings of the back thigh joint in conjunction with fore shin velocity to coarsely differentiate between stance and flight phase. More fine-grained coordination of lift-off and touchdown is handled by the leftmost and rightmost branches, respectively. In addition to periodic neuron activations following regular

gait, we also observe more specialized decision trees that respond to potential safety-critical situations. For example, the decision tree of CMD 3 includes two branching options that align with variations of tripping due to premature touchdown during the flight phase corresponding to a forward trip (6A) and a forward flip (6B). More quantitative analysis is shown in Appendix Section I.

## 4.4 End-to-end Visual Servoing

**Environment and policy learning.** We consider vision-based end-to-end autonomous driving where the neural policy learns steering commands for lane-following. The model takes front-view RGB images of the vehicle as input, and outputs control commands for the steering wheel and speed. We use the high-fidelity data-driven simulator *VISTA* [45] as our environment. We adopt a training strategy called *guided policy learning* that leverages *VISTA* to augment a real-world dataset with diverse synthetic data for robust policy learning. The training dataset contains roughly 200k image-and-control pairs and mean squared error is used as the training objective. For evaluation, we initialize the vehicle at a random position throughout the entire track and evaluate the policy for 100 frames (roughly 10s) for 100 episodes. The performance is estimated as the ratio of the length of the path traversed without a crash and the total path length. Notably, this task has two additional major distinctions: (1) policy learning based on supervised learning as opposed to reinforcement learning (2) policies take in data (images) different from states on which logic program is grounded.

**Quantitative analysis.** In Table 5, we have consistent findings that NCP achieves the best performance in both disentanglement and explanation quality (more precisely, comparable with the best in modularity, horizontal explanation size, and cognitive chunks). In Table 6, the mutual information gap achieves consistently the best alignment in the overall ranking. Also, the correlation are much higher than classical control and comparable with or higher than locomotion. This suggests again that more complex tasks yield better alignment between disentanglement and explanation quality.

**Maneuver strategies from visual inputs.** In Figure 4, we show extracted behaviors for a neuron in the NCP driving policy. While the state space of driving is higher dimensional, we focus on local heading error $\mu$ and lateral deviation from the lane center $d$ in the lane following task. We compute the statistics and plot neuron response and closed-loop dynamics in the $d$-$\mu$ phase portrait. This specific neuron develops more fine-grained control for situations when the vehicle is on the right of the lane center, as shown in Figure 4(a), with images retrieved from neuron response in Figure 4(b).

## 5 Discussion and Limitation

We summarize all consistent findings to answer the questions asked at the beginning of Section 4. First, disentanglement is highly indicative of explanation quality in the best performance across all tasks, suggesting that, among all compact neural policies, the ones with more disentangled representation are more interpretable for humans [41] (with robustness analysis across hyperparameters in Appendix Section F). In addition, compared to neuron response variance and modularity, the mutual information gap consistently has the best alignment in the overall ranking with explanation quality. Besides, there are certain network architectures (NCPs) that exhibit superior performance in disentanglement and explanation quality consistently across experiments. Another interesting finding is that more complex tasks yield better alignment between disentanglement and explanation quality (by comparing between Table 2, 4, and 6). Finally, qualitative results showed that learned behaviors of neural policies, e.g., gait patterns or maneuver strategies, can be extracted from neuron responses.
**Limitation.** The proposed framework involves extracting factor of variation relevant to strategies and task structures; however, the empirical implementation only considers abstraction (i.e., logic program) in a single time step. Extensions to temporal reasoning include temporal logic [46] or using decision trees with temporal capability [33, 47]. Furthermore, the abstraction is grounded on a set of world states pre-determined by human; however, these states may not be sufficiently expressive to capture the learned behavior of the policy. This requires estimating the information carried between observation and grounding symbols or methods to extract the latter from the former.

**Acknowledgments.** This work is supported by Capgemini Engineering, the Toyota Research Institute (TRI). This research was also supported in part by the AI2050 program at Schmidt Futures (Grant G-22-905 63172). It reflects only the opinions of its authors and not TRI or Toyota entity.

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

## A Compact Networks for Neural Policies

To obtain compact neural representations, there are three common approaches: 1) simply choose an RNN with small number of units densely wired to each other (e.g., a long short-term memory, LSTM, network [38], or a continuous-time network such as an ordinary differential equation, ODE, -based network [48, 39]). 2) sparsify a large network into a smaller system (e.g., lottery ticket winners [49], or sparse flows [50]), and 3) use neural circuit policies that are given by sparse architectures with added complexity to their neural and synaptic representations but have a light-weighted network architecture [3, 4, 40].

In the first approach the number of model parameters inversely affect interpretability, i.e., interpreting wider and/or deeper densely wired RNNs exponentially makes the interpretation of the system harder. Sparsity has been shown to help obtain a network with 95% less parameters compared to the initial model. However, recent studies show that such levels of sparsity affect the robustness of the model, thus make it more susceptible to perturbations [51]. Neural circuit policies (NCPs) [4] on the other hand have shown great promise in achieving attractive degrees of generalizability while maintaining robustness to environmental perturbations. This representation learning capability is rooted in their ability to capture the true cause and effect of a given task [5]. NCPs are sparse network architectures with their nodes and edges determined by a liquid time-constant (LTC) concept [3]. The state of a liquid network is described by the following set of ODEs [3]: $\frac{d\mathbf{x}(t)}{dt} = -\left[\frac{1}{\tau} + f(\mathbf{x}(t), \mathbf{I}(t), t, \theta)\right] \odot \mathbf{x}(t) + f(\mathbf{x}(t), \mathbf{I}(t), t, \theta) \odot A$. Here, $\mathbf{x}^{(D \times 1)}(t)$ is the hidden state with size D, $\mathbf{I}^{(m \times 1)}(t)$ is an input signal, $\tau^{(D \times 1)}$ is the fixed internal time-constant vector, $A^{(D \times 1)}$ is a bias parameter, and $\odot$ is the Hadamard product. In tasks involving spatiotemporal dynamics these networks showed significant benefit over their counterparts, both in their ODE form and in their closed-form representation termed Closed-form continuous-time (CfC) models [4, 5, 40].

**Interpretation of Neuron Responses.** Compact neural representations promise to enable the interpretability of decision-making by focusing post-hoc analysis on a limited number of neural responses. However, having merely a lower-dimensional space for visualization is not sufficient to identify consistent behaviors or strategies acquired by a learning agent. Emergent behaviors may distribute responses across numerous neurons with a high degree of entanglement. Even for models with a small number of neurons, it can be challenging to identify and interpret the behavior correlated with observed response patterns. In this paper, we hypothesize that abstraction with respect to a type of learned strategy within a single neuron is necessary for better interpretability of neural policies. We further desire semantic grounding of the neuron response, that is, associating neuron response to human-readable representation. The representation space should be abstract enough to be human-understandable and expressive enough to capture arbitrary types of emergent behaviors or strategies. We adopt the framework of logic programs due to their simple yet effective representations of decision-making processes.

## B A Motivating Perspective Of Disentangled Representation

The underlying behaviors of neural policies involves descriptions with multiple levels of abstraction, from detailed states at every time instance to high-level strategies toward solving a task, spanning a continuum where the details can be summarized and reduced to gradually construct their concise counterparts. Among these descriptions of behaviors, a right amount of abstraction should be concise enough for human interpretability yet being sufficiently informative of how neural policies act locally toward solving the overall task. Relevant concepts about abstraction have been explored in the context of state abstraction in Markov Decision Process (MDP) [52], hierarchical reinforcement learning [53], and developmental psychology [54]. In the following, we aim to more formally define such abstraction for interpretability of neural policies and draw connection to disentangled representations. First, we define a MDP as a tuple $\{\mathcal{S}, \mathcal{A}, P_a, R\}$, where at time instance $t$, $s_t \in \mathcal{S}$ is the state, $a_t \in \mathcal{A}$ is the action, $P_a : \mathcal{S} \times \mathcal{A} \times \mathcal{S} \to [0, 1]$ is the transition function, $R : \mathcal{S} \times \mathcal{A} \to \mathbb{R}$ is the reward function. The goal of policy learning in a MDP is to find a policy $\pi : \mathcal{S} \times \mathcal{A} \to [0, 1]$

that maximizes the expected future return (accumulated reward). The closed-loop dynamics (in deterministic setting) can then be written as

$$s_{t+1} = P_a(s_t, a_t), \quad \text{where } a_t = \pi(s_t)$$

Then, we construct an abstract MDP, with state presumably being the abstraction we are looking for interpretability, as a tuple $\{\hat{S}, \hat{A}, \hat{P}_a, \hat{R}\}$ that follows similar definition to the above-mentioned regular MDP. It follows the deterministic MDP homomorphism [55, 56] as follows,

$$\forall s_t, s_{t+1} \in S, a_t \in A \quad P_a(s_t, a_t) = s_{t+1} \Rightarrow \bar{P}_a(Q(s_t), \bar{A}(a_t)) = Q(s_{t+1})$$
$$\forall s_t \in S, a_t \in A \quad R(s_t, a_t) = R(Q(s_t), \bar{A}(a_t))$$

where $Q : S \rightarrow \hat{S}$ is the state embedding function and $\bar{A} : A \rightarrow \hat{A}$ is the action embedding function. The state embedding function can also be seen as an action-equivariant map that precisely satisfies the MDP homomorphism [56]. Next, we start to draw connection to disentangled representation from one of its formalism using symmetries and group theory [57]. Informally, disentanglement refers to the level of decomposition in representation that reflects the *factor of variation*. For example, one dimension of vector representations corresponds to color and the other corresponds to shape. In [57], these factor of variations are formally defined as symmetries of world state ($S$ in our case). Given group $G$, binary operator $\circ : G \times G \rightarrow G$, group decomposition into a direct product of subgroups $G = G_1 \times G_2 \times \ldots$, and group action $\cdot_X : G \times X \rightarrow X$ with $X$ as a set which the group action act upon, the idea is to "commute" symmetries from one set $X$ to the other $X'$. Suppose there is a group $G$ of geometries acting on the world state $S$ via action $\cdot_S : G \times S \rightarrow S$, we would like to find a corresponding action acting on representation $\cdot_Z : G \times Z \rightarrow Z$ that reflects the symmetric structure of $S$ in $Z$ (in our case neuron response $z_t \in Z$). This entails the equivariance condition,

$$g \cdot_Z E_{S \rightarrow Z}(s_t) = E_{S \rightarrow Z}(g \cdot_S s_t)$$

where $E_{S \rightarrow Z}$ commutes action across $S$ and $Z$, and can be called a G-morphism or equivariant map.

$$
\begin{array}{ccc}
G \times S & \xrightarrow{\cdot_S} & S \\
{\scriptstyle id_G \times E_{S \rightarrow Z}} \downarrow & & \downarrow {\scriptstyle E_{S \rightarrow Z}} \\
G \times Z & \dashrightarrow{\cdot_Z} & Z
\end{array}
$$

A more concrete connection of group action to MDP can be seen in the analogy of agent-environment interaction [58],

$$g \cdot_S s_t = s_{t+1} = P_a(s_t, a_t)$$

It is worth emphasizing the distinction of group action $\cdot_S$ and regular action $a_t$: not all regular action $a_t$ exhibit symmetry, as pointed out in [58]. And the group action upon neural state $\cdot_Z$ can be viewed as the transition dynamics of neural policies,

$$g \cdot_Z z_t = z_{t+1} = \pi_z(z_t) = \pi_s(P_a(s_t, \pi_a(z_t)))$$

where $\pi = \pi_a \circ \pi_s, \pi_a : Z \rightarrow A, \pi_s : S \rightarrow Z$ is simply the decomposition of neural policies to explicitly extract neuron responses $z_t$ and $\pi_z : Z \rightarrow Z$ is the transition function of neural states (note that this does not necessarily require recurrence structure of neural policies; instead this is more of a convenient notation here). Following the definition of [57], an agent's representation $Z$ is disentangled with respect to the decomposition $G = G_1 \times G_2 \times \ldots$ if

1. There is a group action $\cdot_Z : G \times Z \rightarrow Z$.

2. The map $E_{S \rightarrow Z} : S \rightarrow Z$ is equivariant between the group actions on $S$ and $Z$.

3. There is a decomposition $Z = Z_1 \times Z_2 \times \ldots$ such that each $Z_i$ is fixed by the actions of all $G_j, j \neq i$ and affected only by $G_i$.

For the first condition, We already define $\cdot_Z$ in the above. For the second condition, we show that the equivariant map can follow the definition $E_{S \rightarrow Z} = \pi_z \circ \pi_s$, i.e., $z_{t+1} = E_{S \rightarrow Z}(s_t)$. This follows the proof as,

$$g \cdot_Z E_{S \rightarrow Z}(s_t) = g \cdot_Z z_{t+1} = \pi_z(z_{t+1}) = \pi_z(\pi_s(s_{t+1})) = (\pi_z \circ \pi_s)(s_{t+1}) = E_{S \rightarrow Z}(g \cdot_S s_t)$$

Next, extending the formalism of disentangled representation in [57] with the above-mentioned MDP homomorphism [55], we define the equivariance condition between the regular MDP $\{\mathcal{S}, \mathcal{A}, P_a, R\}$ and the abstract MDP $\{\hat{\mathcal{S}}, \hat{\mathcal{A}}, \hat{P}_a, \hat{R}\}$,

$$g \cdot_{\mathcal{S}} E_{\hat{\mathcal{S}} \to \mathcal{S}}(\hat{s}_t) = E_{\hat{\mathcal{S}} \to \mathcal{S}}(g \cdot_{\hat{\mathcal{S}}} \hat{s}_t)$$

where $E_{\hat{\mathcal{S}} \to \mathcal{S}}$ commutes action across $\hat{\mathcal{S}}$ and $\mathcal{S}$, and can be defined with MDP homomorphism,

$$\hat{s}_{t+1} = Q(s_{t+1})$$
$$\hat{P}_a(\hat{s}_t, \hat{a}_t) = Q(P_a(s_t, a_t))$$
$$g \cdot_{\hat{\mathcal{S}}} \hat{s}_t = Q(g \cdot_{\mathcal{S}} s_t)$$
$$Q^{-1}(g \cdot_{\hat{\mathcal{S}}} \hat{s}_t) = g \cdot_{\mathcal{S}} Q^{-1}(\hat{s}_t)$$
$$E_{\hat{\mathcal{S}} \to \mathcal{S}} = Q^{-1}$$

Note that theoretically the state embedding function $Q$ may not have an inverse mapping since going from $\mathcal{S}$ to $\hat{\mathcal{S}}$ is supposed to be more abstract (and thus concise with equal or less information). However, this does not matter since we don't necessarily require this recipe to tell us how exactly group actions in $\hat{\mathcal{S}}$ commute to $\mathcal{S}$. Overall, we establish the following group homomorphism across set $\hat{\mathcal{S}}$, $\mathcal{S}$, and $Z$,

$$
\begin{array}{ccc}
G \times \hat{\mathcal{S}} & \xrightarrow{\cdot_{\hat{\mathcal{S}}}} & \hat{\mathcal{S}} \\
{\scriptstyle id_G \times E_{\hat{\mathcal{S}} \to \mathcal{S}}} \downarrow & & \downarrow {\scriptstyle E_{\hat{\mathcal{S}} \to \mathcal{S}}} \\
G \times \mathcal{S} & \dashrightarrow{\cdot_{\mathcal{S}}} & \mathcal{S} \\
{\scriptstyle id_G \times E_{\mathcal{S} \to Z}} \downarrow & & \downarrow {\scriptstyle E_{\mathcal{S} \to Z}} \\
G \times Z & \dashrightarrow{\cdot_{Z}} & Z
\end{array}
$$

This connects *the right amount of abstraction for interpretability* discussed in the beginning, then associated with MDP homomorphism, to *factor of variation* in disentangled representation, which is formalized by symmetry and group theory. Disentanglement in $Z$ can then be lifted to symmetries in abstract state space $\hat{\mathcal{S}}$. In [57], disentanglement of representation is lifted up to the symmetries in the world state space $\mathcal{S}$, e.g., a factor of group decomposition $G_i$ can be color of an object. However, this is not sufficient to describe the behavior of policies since $\mathcal{S}$ lacks task structure. Hence, we further go from $\mathcal{S}$ to $\hat{\mathcal{S}}$ with MDP homomorphism to capture the essence of solving a task. The factor of group decomposition $G_i$ can then be task-related, e.g., relative pose to a target object (which may be of high interest for tasks like object tracjing, and less so for tasks like locomotion). Overall, this provides a motivation to cast the problem of searching for proper description of the behavior of neural policies (for interpretability) to searching for disentanglement in neuron responses. In this paper, we therefore study how to measure interpretability of compact neural policies with disentangled representation.

## C  Calibration Of Mutual Information

**Lemma C.1.** *The calibration term* $I[z^j; \mathcal{P}_k] - I[z^j; \mathcal{P}_k; \mathcal{P}_{k^j}]$ *in both MIG (5) and Modularity (6) metrics, for $j \neq i^*$, without loss of generality has the following lower bound:*

$$I[z^j; \mathcal{P}_k] - I[z^j; \mathcal{P}_k; \mathcal{P}_{k^j}] \geq \max(0,\ I[z^j; \mathcal{P}_k] - I[\mathcal{P}_k; \mathcal{P}_{k^j}]) \tag{7}$$

Lemma C.1 is necessary because to compute the calibration term, we need access to the conditional distribution of the random variable $(\mathcal{P}_{k^j} | z^j)$, which is normally inaccessible. Hence, we derive a lower bound for the calibrated mutual information.

*Proof.* In the main paper, we adapting Mutual Information Gap (MIG) [21] to our framework as,

$$\frac{1}{K} \sum_{k=1}^{K} \frac{1}{H[\mathcal{P}_k]} \left( I[z^{i^*}; \mathcal{P}_k] - \max_{j \neq i^*} I[z^j; \mathcal{P}_k] - I[z^j; \mathcal{P}_k; \mathcal{P}_{k^j}] \right)$$

and Modularity score [36] as,

$$\frac{1}{\mathcal{I}} \sum_{i \in \mathcal{I}} 1 - \frac{\sum_{k \neq k^*} (I[z^i; \mathcal{P}_k] - I[z^i; \mathcal{P}_k; \mathcal{P}_{k^*}])^2}{(K-1) I[z^i; \mathcal{P}_{k^*}]^2}$$

Both involve the computation of $I[z^j; \mathcal{P}_k; \mathcal{P}_{k^j}]$. Without loss of generality for both cases (and with the notation of MIG), we simplify the calibration term for $j \neq i^*$ as follows,

$$
\begin{aligned}
& I[z^j; \mathcal{P}_k] - I[z^j; \mathcal{P}_k; \mathcal{P}_{k^j}] \\
= {}& I[z^j; \mathcal{P}_k] - (I[z^j; \mathcal{P}_k] - I[z^j; \mathcal{P}_k | \mathcal{P}_{k^j}]) \\
= {}& I[z^j; \mathcal{P}_k | \mathcal{P}_{k^j}] \\
= {}& I[z^j; \mathcal{P}_k] + H[\mathcal{P}_{k^j} | z^j] + H[\mathcal{P}_{k^j} | \mathcal{P}_k] - H[\mathcal{P}_{k^j} | z^j, \mathcal{P}_k] - H[\mathcal{P}_{k^j}] \\
= {}& I[z^j; \mathcal{P}_k] - (H[\mathcal{P}_{k^j}] - H[\mathcal{P}_{k^j} | \mathcal{P}_k]) + (H[\mathcal{P}_{k^j} | z^j] - H[\mathcal{P}_{k^j} | z^j, \mathcal{P}_k]) \\
= {}& I[z^j; \mathcal{P}_k] - I[\mathcal{P}_k; \mathcal{P}_{k^j}] + I[\mathcal{P}_{k^j} | z^j; \mathcal{P}_k] \\
\geq {}& \max(0, \ I[z^j; \mathcal{P}_k] - I[\mathcal{P}_k; \mathcal{P}_{k^j}])
\end{aligned}
$$

Most steps simply follow identities of mutual information and entropy. The last step requires access to the conditional distribution of random variable $(\mathcal{P}_{k^j} | z^j)$, which is normally inaccessible. Hence, we introduce an approximation that serves as a lower bound for the calibrated mutual information in our implementation. $\qquad \square$

## D  Other Quantitative Measures

**Decision Path Accuracy.** During deployment, we use an inverse proxy $q_{\phi^i}$ for the decision tree $T_{\theta^i}$ and hence we compute the approximation error by measuring the accuracy of a state-grounded decision path inferred from the neuron response with $q_{\phi^i}$ compared to true states,

$$\frac{1}{|\mathcal{I}|} \sum_{i \in \mathcal{I}} \frac{1}{|\mathcal{D}_{\mathrm{dt}}|} \sum_{(s_t, z_t^i) \in \mathcal{D}_{\mathrm{dt}}} \frac{1}{|q_{\phi^i}(z_t^i)|} \sum_{\substack{n \in q_{\phi^i}(z_t^i) \\ j = g(n)}} \mathbb{1}[s_t^j \leq c_n] \tag{8}$$

where $\mathbb{1}$ is an indicator function, $q_{\phi^i}(z_t^i)$ is the inferred decision path with norm as number of decision rules. The condition $s_t^j \leq c_n$ validates if the current state $s_t^j$ complies with the inferred rule defined by $c_n$ (which is from $T_{\theta^i}$). Since the discrepancy is computed at the decision rule level, it captures not only the error of the classifier model $q_{\phi^i}$ but also how accurately $f_{\mathcal{S}}^i$ parses $z_t^i$.

**Cross-neuron Logic Conflict.** When interpreting a neural policy as a whole instead of inspecting individual neuron response, it is straightforward to find the intersection across logic programs extracted from different neurons $l_t = \texttt{reduce}(\wedge_{i \in \mathcal{I}} l_t^i)$, where $\texttt{reduce}$ summarizes and reduces logic programs to a more compact one. Intuitively, the neuron-wise logic program should summarize the operational domain of the strategy currently executed by the neuron, where intersection describes the domain of a joint strategy across neurons. However, the reduction of intersection can be invalid if there is conflict in the logical formulae across neurons, e.g., $a \leq 3$ from the first neuron and $a \geq 4$ from the second neuron. The conflict may imply, under the same configuration of $f_{\mathcal{S}}$, that (1) the policy fails to learn compatible strategies across neurons or (2) there is an error induced by the interpreter due to insufficient or ambiguous connection between the logic program and the neuron response, which implicitly indicates lack of interpretability.

**Experimental Results.** For classical control, we verify in Table 7 that all models achieve comparable performance when learning toward target -500 episode reward. For locomotion, in Table 8, most models achieve comparable task performance except for GRU and ODE-RNN being slightly worse. For end-to-end visual servoing, in Table 9, all models achieve good performance ($> 0.9$) except for ODE-RNN, which fails to learn a good policy within maximal training iterations.

Table 7: Other quantitative results of classical control.

| Network Architecture | Decision Path Accuracy ↑ | Logic Conflict ↓ | Performance ↑ |
|---|---|---|---|
| FCs | $0.3015^{.069}$ | $0.2104^{.065}$ | $-488.55^{010.99}$ |
| GRU | $0.2504^{.104}$ | $0.2832^{.080}$ | $-559.82^{114.07}$ |
| LSTM | $0.2392^{.031}$ | $0.5072^{.103}$ | $\mathbf{-467.95}^{024.53}$ |
| ODE-RNN | $0.2980^{.065}$ | $0.2506^{.101}$ | $-533.93^{122.47}$ |
| CfC | $0.2509^{.138}$ | $\mathbf{0.1556}^{.099}$ | $-489.28^{007.66}$ |
| NCP | $\mathbf{0.4726}^{.114}$ | $0.2026^{.088}$ | $-556.64^{116.75}$ |

Table 8: Other quantitative results of locomotion.

| Network Architecture | Decision Path Accuracy ↑ | Logic Conflict ↓ | Performance ↑ |
|---|---|---|---|
| FCs | $0.5285^{.054}$ | $\mathbf{0.1035}^{.011}$ | $5186.50^{2458.84}$ |
| GRU | $0.4924^{.054}$ | $0.1500^{.032}$ | $3857.21^{1448.57}$ |
| LSTM | $0.5283^{.073}$ | $0.2155^{.042}$ | $4122.74^{1751.04}$ |
| ODE-RNN | $0.4959^{.057}$ | $0.1474^{.024}$ | $3472.69^{1734.91}$ |
| CfC | $0.4841^{.045}$ | $0.1581^{.031}$ | $5195.46^{2292.67}$ |
| NCP | $\mathbf{0.5859}^{.019}$ | $0.1105^{.026}$ | $\mathbf{5822.73}^{0512.73}$ |

# E    Implementation Details

NCPs are designed by a four-layer structure consisting of sensory neurons (input layer), interneurons, command neurons (with recurrent connections), and motor neurons (output layer). To make a fair comparison, we augment all non-NCP models by a feed-forward layer, which is of equivalent size to the inter-neuron layer in NCPs.

## E.1    Classical Control (Pendulum)

**Network Architecture.** With 3-dimensional observation space and 1-dimensional action space, we use the following network architecture for compact neural policies.

- *FCs*: a $3 \rightarrow 10 \rightarrow 4 \rightarrow 1$ fully-connected network with *tanh* activation.

- *GRU*: a $3 \rightarrow 10$ fully-connected network with *tanh* activation followed by GRU with cell size of 4, outputting a 1-dimensional action.

- *LSTM*: a $3 \rightarrow 10$ fully-connected network with *tanh* activation followed by LSTM with hidden size of 4, outputting a 1-dimensional action. Note that this effectively gives 8 cells by considering hidden and cell states.

- *ODE-RNN*: a $3 \rightarrow 10$ fully-connected network with *tanh* activation followed by a neural ODE with recurrent component both of size 4, outputting a 1-dimensional action.

- *CfC*: with backbone layer $= 1$, backbone unit $= 10$, backbone activation *silu*, hidden size $= 4$ without gate and mixed memory, outputting a 1-dimensional action.

- *NCP*: with 3 sensory neurons, 10 interneuron, 4 command neurons, 1 motor neuron, 4 output sensory synapses, 3 output inter-synapses, 2 recurrent command synapse, 3 motor synapses.

For all policies, we use a $3 \rightarrow 64 \rightarrow 64 \rightarrow 1$ fully-connected networks with *tanh* activation as value function. We interpret the layer of size 4 for each policy.

**Training details.** We use PPO with the following parameters for all models. Learning rate is $0.0003$. Train batch size (of an epoch) is $512$. Mini-batch size is $64$. Number of iteration within a batch is $6$. Value function clip parameter is $10.0$. Discount factor of the MDP is $0.95$. Generalized advantage

Table 9: Other quantitative results of visual servoing.

| Network Architecture | Decision Path Accuracy ↑ | Logic Conflict ↓ | Performance ↑ |
|---|---|---|---|
| FCs | 0.5379 | 0.1354 | **1.0000** |
| GRU | **0.6160** | 0.1884 | 0.9210 |
| LSTM | 0.5174 | 0.4504 | **1.0000** |
| ODE-RNN | 0.5483 | 0.3786 | 0.4239 |
| CfC | 0.5549 | 0.2274 | 0.9922 |
| NCP | 0.5960 | **0.1067** | **1.0000** |

estimation parameter is 0.95. Initial coefficient of KL divergence is 0.2. Clip parameter is 0.3. Training halts if reaching target average episode reward 150. Maximal training steps is 1M.

**Interpreter details.** For the decision tree $T_{\theta^i}$, we set minimum number of samples required to be at a leaf node as 10% of the training data, criterion of a split as mean squared error with Friedman's improvement score, the maximum depth of the tree as 3, complexity parameter used for minimal cost-complexity pruning as 0.003; we use scikit-learn implementation of CART (Classification and Regression Trees). For simplicity, we use another decision tree as decision path classifier $q_{\phi^i}$ with maximal depth of tree as 3, minimum number of samples in a leaf node as 1% of data, complexity parameter for pruning as 0.01, criterion as Gini impurity. The state grounding $\mathcal{S}$ of the interpreter $f_{\mathcal{S}}^i$ is $\{\theta, \dot{\theta}\}$, where $\theta$ is joint angle and $\dot{\theta}$ is joint angular velocity. We use the offline data collected during the closed-loop policy evaluation for the training dataset, which consists of 100 trajectories with each having maximally 100 time steps (default in the environment).

### E.2 Locomotion (HalfCheetah)

**Network Architecture.** With 17-dimensional observation space and 6-dimensional action space, we first use feature extractors of a shared architecture as a $17 \rightarrow 256$ fully-connected network, which then output features to compact neural policies with the following architectures,

- *FCs*: a $256 \rightarrow 20 \rightarrow 10 \rightarrow 6$ fully-connected network with *tanh* activation.

- *GRU*: a $256 \rightarrow 20$ fully-connected network with *tanh* activation followed by GRU with cell size of 10, outputting a 6-dimensional action.

- *LSTM*: a $256 \rightarrow 20$ fully-connected network with *tanh* activation followed by LSTM with hidden size of 10, outputting a 6-dimensional action. Note that this effectively gives 20 cells by considering hidden and cell states.

- *ODE-RNN*: a $256 \rightarrow 20$ fully-connected network with *tanh* activation followed by a neural ODE with recurrent component both of size 10, outputting a 6-dimensional action.

- *CfC*: with backbone layer $= 1$, backbone unit $= 20$, backbone activation *silu*, hidden size $= 10$ without gate and mixed memory.

- *NCP*: with 256 sensory neurons, 20 interneuron, 10 command neurons, 6 motor neuron, 4 output sensory synapses, 5 output inter-synapses, 6 recurrent command synapse, 4 input motor synapses.

For all policies, we use a $17 \rightarrow 256 \rightarrow 256 \rightarrow 1$ fully-connected networks with *tanh* activation as value function. We interpret the layer of size 10 for each policy.

**Training details.** We use PPO with the following parameters for all models. Learning rate is 0.0003. Train batch size (of an epoch) is 65536. Mini-batch size is 4096. Number of iteration within a batch is 32. Value function coefficient is 10.0. Discount factor of the MDP is 0.99. Generalized advantage estimation parameter is 0.95. Initial coefficient of KL divergence is 1.0. Clip parameter is 0.2. Gradient norm clip is 0.5. Training halts if reaching target average episode reward $-500$. Maximal training steps is 12M.

**Interpreter details.** For the decision tree $T_{\theta^i}$, we set minimum number of samples required to be at a leaf node as $10\%$ of the training data, criterion of a split as mean squared error with Friedman's improvement score, the maximum depth of the tree as 3, complexity parameter used for minimal cost-complexity pruning as $0.001$; we use scikit-learn implementation of CART (Classification and Regression Trees). For simplicity, we use another decision tree as decision path classifier $q_{\phi^i}$ with maximal depth of tree as 3, minimum number of samples in a leaf node as $1\%$ of data, complexity parameter for pruning as $0.01$, criterion as Gini impurity. The state grounding $\mathcal{S}$ of the interpreter $f_{\mathcal{S}}^i$ is $\{h_R, \theta_R, \theta_{T,B}, \theta_{S,B}, \theta_{F,B}, \theta_{T,F}, \theta_{S,F}, \theta_{F,F}, \dot{x}_R, \dot{h}_R, \dot{\theta}_R, \dot{\theta}_{T,B}, \dot{\theta}_{S,B}, \dot{\theta}_{F,B}, \dot{\theta}_{T,F}, \dot{\theta}_{S,F}, \dot{\theta}_{F,F}\}$, where $h_R, \dot{h}_R$ are position and velocity of z-coordinate of the front tip, $\theta_R, \dot{\theta}_R$ are angle and angular velocity of the front tip, $\theta_{T,B}, \dot{\theta}_{T,B}$ are angle and angular velocity of the thigh in the back, $\theta_{S,B}, \dot{\theta}_{S,B}$ are angle and angular velocity of the shin in the back, $\theta_{F,B}, \dot{\theta}_{F,B}$ are angle and angular velocity of the foot in the back, $\theta_{T,T}, \dot{\theta}_{T,T}$ are angle and angular velocity of the thigh in the front, $\theta_{S,T}, \dot{\theta}_{S,T}$ are angle and angular velocity of the shin in the front, $\theta_{F,T}, \dot{\theta}_{F,T}$ are angle and angular velocity of the foot in the front, $\dot{x}_R$ is the velocity of x-coordinate of the front tip. We use the offline data collected during the closed-loop policy evaluation for the training dataset, which consists of 100 trajectories with each having maximally 1000 time steps (default in the environment).

### E.3 End-to-end visual servoing (Image-based Driving)

**Network Architecture.** With image observation space of size $(200, 320, 3)$ and 2-dimensional action space, we first use feature extractors of a shared architecture as a convolutional neural network (CNN) in Table 10, which then output features to compact neural policies with the following architectures,

- *FCs*: a $1280 \rightarrow 20 \rightarrow 8 \rightarrow 2$ fully-connected network with *tanh* activation.
- *GRU*: a $1280 \rightarrow 20$ fully-connected network with *tanh* activation followed by GRU with cell size of 8, outputting a 2-dimensional action.
- *LSTM*: a $1280 \rightarrow 20$ fully-connected network with *tanh* activation followed by LSTM with hidden size of 8, outputting a 2-dimensional action. Note that this effectively gives 20 cells by considering hidden and cell states.
- *ODE-RNN*: a $1280 \rightarrow 20$ fully-connected network with *tanh* activation followed by a neural ODE with recurrent component both of size 8, outputting a 2-dimensional action.
- *CfC*: with backbone layer $= 1$, backbone unit $= 20$, backbone activation *silu*, hidden size $= 8$ without gate and mixed memory.
- *NCP*: with 1280 sensory neurons, 20 interneuron, 8 command neurons, 2 motor neuron, 4 output sensory synapses, 5 output inter-synapses, 6 recurrent command synapse, 4 input motor synapses.

**Training details.** Batch size is 64. Sequence size is 10. Learning rate is 0.001. Number of epochs is 10. We perform data augmentation on RGB images with randomized gamma of range $[0.5, 1.5]$, brightness of range $[0.5, 1.5]$, contrast of range $[0.7, 1.3]$, saturation of range $[0.5, 1.5]$.

**Interpreter details.** For the decision tree $T_{\theta^i}$, we set minimum number of samples required to be at a leaf node as $10\%$ of the training data, criterion of a split as mean squared error with Friedman's improvement score, the maximum depth of the tree as 3, complexity parameter used for minimal cost-complexity pruning as $0.003$; we use scikit-learn implementation of CART (Classification and Regression Trees). For simplicity, we use another decision tree as decision path classifier $q_{\phi^i}$ with maximal depth of tree as 3, minimum number of samples in a leaf node as $1\%$ of data, complexity parameter for pruning as $0.01$, criterion as Gini impurity. The state grounding $\mathcal{S}$ of the interpreter $f_{\mathcal{S}}^i$ is $\{v, \delta, d, \Delta l, \mu, \kappa\}$, where $v$ is vehicle speed, $\delta$ is heading, $d$ is lateral deviation from the lane center, $\Delta l$ is longtitudinal deviation from the lane center, $\mu$ is local heading error with respect to the lane center, $\kappa$ is road curvature. We use the offline data collected during the closed-loop policy evaluation for the training dataset, which consists of 100 trajectories with each having maximally 100 time steps.

| Layer | Hyperparameters |
|-------|-----------------|
| Conv2d | (3, 24, 5, 2, 2) |
| GroupNorm2d | (16, 1e-5) |
| ELU | - |
| Dropout | 0.3 |
| Conv2d | (24, 36, 5, 2, 2) |
| GroupNorm2d | (16, 1e-5) |
| ELU | - |
| Dropout | 0.3 |
| Conv2d | (36, 48, 3, 2, 1) |
| GroupNorm2d | (16, 1e-5) |
| ELU | - |
| Dropout | 0.3 |
| Conv2d | (48, 64, 3, 1, 1) |
| GroupNorm2d | (16, 1e-5) |
| ELU | - |
| Dropout | 0.3 |
| Conv2d | (64, 64, 3, 1, 1) |
| AdaptiveAvgPool2d | reduce height dimension |

Table 10: Network architecture of CNN feature extractor for end-to-end visual servoing. Hyperparameters for *Conv2d* are input channel, output channel, kernel size, stride, and padding; for *GroupNorm2d*, they are group size and epsilon; for *Dropout*, it is drop probability.

# F   Robustness Analysis

We propose to study the interpretability of neural policies through decision trees and present several quantitative measures of interpretability by analyzing various properties on top of neuron responses and corresponding decision trees, including *Neural-Response Variance*, *Mutual Information Gap*, *Modularity*, *Decision Path Accuracy*, and *Logic Conflict*. However, the extracted decision trees may differ across different configurations. Hence, to validate the robustness of the proposed metrics to hyperparameters, we compute all metrics with different decision tree parameters in classical control environment (Pendulum). We report the averaged results with 5 random seeds in Table 11 (*Neural-Response Variance*), Table 12 (*Mutal Information Gap*), Table 13 (*Modularity*), Table 14 (*Decision Path Accuracy*), Table 15 (*Logic Conflict*). Most metrics (variance, MI-gap, decision path accuracy, logic conflict) yield consistent top-1 results and agree with similar rankings among network architectures, except for modularity that is slightly less robust against hyperparameters yet still consistent in the top-3 set of models. This results demonstrate the reliability of the proposed interpretability analysis for neural policies.

Table 11: Robustness to hyperparameters for *Neural-Response Variance*. The results are averaged across 5 random seeds in classical control (Pendulum).

| [Variance ↓] Network Architecture | | FCs | GRU | LSTM | ODE-RNN | CfC | NCP |
|-----------------------------------|------|--------|--------|------------|---------|--------|--------|
| Cost Complexity Pruning | 0.001 | 0.0232 | 0.0304 | 0.0209 | 0.0266 | 0.0254 | **0.0207** |
| | 0.003 | 0.0242 | 0.0329 | **0.0216** | 0.0287 | 0.0272 | 0.0240 |
| | 0.01 | 0.0261 | 0.0371 | **0.0221** | 0.0315 | 0.0267 | 0.0305 |
| Minimal Leaf Sample Ratio | 0.01 | 0.0154 | 0.0261 | **0.0138** | 0.0193 | 0.0189 | 0.0186 |
| | 0.1 | 0.0242 | 0.0329 | **0.0216** | 0.0287 | 0.0272 | 0.0240 |
| | 0.2 | 0.0334 | 0.0387 | **0.0284** | 0.0354 | 0.0295 | 0.0285 |

Table 12: Robustness to hyperparameters for *Mutual Information Gap*. The results are averaged across 5 random seeds in classical control (Pendulum).

| [MI-Gap ↑] Network Architecture | | FCs | GRU | LSTM | ODE-RNN | CfC | NCP |
|---|---|---|---|---|---|---|---|
| Cost Complexity Pruning | 0.001 | 0.0284 | 0.2686 | 0.2026 | 0.2891 | 0.2544 | **0.3403** |
| | 0.003 | 0.3008 | 0.2764 | 0.2303 | 0.3062 | 0.2892 | **0.3653** |
| | 0.01 | 0.3482 | 0.3065 | 0.2547 | 0.3142 | 0.3567 | **0.3664** |
| Minimal Leaf Sample Ratio | 0.01 | 0.2824 | 0.2632 | 0.2040 | 0.2819 | 0.2433 | **0.3456** |
| | 0.1 | 0.3008 | 0.2764 | 0.2303 | 0.3062 | 0.2892 | **0.3653** |
| | 0.2 | **0.3798** | 0.3387 | 0.2528 | 0.3168 | 0.3342 | 0.3429 |

Table 13: Robustness to hyperparameters for *Modularity*. The results are averaged across 5 random seeds in classical control (Pendulum).

| [Modularity ↑] Network Architecture | | FCs | GRU | LSTM | ODE-RNN | CfC | NCP |
|---|---|---|---|---|---|---|---|
| Cost Complexity Pruning | 0.001 | **0.9519** | 0.9558 | 0.9327 | 0.9485 | 0.9228 | 0.9438 |
| | 0.003 | 0.9471 | 0.9550 | 0.9402 | 0.9486 | 0.9116 | **0.9551** |
| | 0.01 | 0.9532 | **0.9598** | 0.9445 | 0.9487 | 0.8970 | 0.9593 |
| Minimal Leaf Sample Ratio | 0.01 | 0.9638 | **0.9702** | 0.9547 | 0.9630 | 0.9333 | 0.9651 |
| | 0.1 | 0.9471 | 0.9550 | 0.9402 | 0.9486 | 0.9116 | **0.9551** |
| | 0.2 | **0.9475** | 0.9372 | 0.9197 | 0.9404 | 0.8755 | 0.9301 |

## G   Counterfactual Analysis via Removal of Neurons

There exist some neurons with logic programs that are sensible but may have little effect on task performance. For example, in NCPs (not confined to this specific architecture but just focus on it for discussion), we find a neuron that aligns its response purely with vehicle speed. Given the task objective is lane following without crashing, such neuron pays attention to useful (for temporal reasoning across frames) but relatively unnecessary (to the task) information. Furthermore, there are neurons that don't exhibit sufficient correlation with any of the environment state and fail to induce decision branching. In light of these observation, we try to remove neurons that we suspect to have little influence on the performance by inspecting their logic program. We show the results in Table 16. Removing neurons 3, 4, 7 has a marginal impact on task performance. Among them, neuron 3 and 4 mostly depends on vehicle speed $v$ with a small tendency to the lateral deviation $d$. Neuron 7 fails to split a tree.

## H   Interpretation Of Driving Maneuver

In Figure 4, we describe interpretations similar to classical control (for a neuron in NCP). While the state space of driving is higher dimensional (5 with bicycle model for lane following), states of interest only include local heading error $\mu$ and lateral deviation from the lane center $d$ in lane following task. We compute the statistics and plot neuron response and closed-loop dynamics in the $d$-$\mu$ phase portrait. This specific neuron develops more fine-grained control for situations when the vehicle is on the right of the lane center, as shown in Figure 4(a). We further show front-view images retrieved based on neuron response in Figure 4(b).

## I   Additional Quantitative Analysis on Locomotion Behaviors

In Figure 3 (right), we demonstrate interesting qualitative examples on discovered gaits critical for failure modes like early touchdown or forward flipping. In Table 17, we conduct quantitative analysis to further justify our findings. Since Gym Half-Cheetah does not have early termination, we did analysis on the stepwise reward, specifically the run reward (horizontal distance incremented across the consecutive time steps). Recall in Section 3.1 paragraph "From neuron responses to decision paths", we can infer the decision path (or branch) by the range of neuron responses. We use

Table 14: Robustness to hyperparameters for *Decision Path Accuracy*. The results are averaged across 5 random seeds in classical control (Pendulum).

| [Decision Path Accuracy ↑]
Network Architecture | | FCs | GRU | LSTM | ODE-RNN | CfC | NCP |
|---|---|---|---|---|---|---|---|
| Cost Complexity Pruning | 0.001 | 0.2815 | 0.2415 | 0.2195 | 0.2904 | 0.2250 | **0.4294** |
| | 0.003 | 0.3015 | 0.2504 | 0.2392 | 0.2980 | 0.2509 | **0.4726** |
| | 0.01 | 0.3074 | 0.3330 | 0.3161 | 0.3707 | 0.2864 | **0.4390** |
| Minimal Leaf Sample Ratio | 0.01 | 0.2950 | 0.2637 | 0.2270 | 0.2574 | 0.2452 | **0.4287** |
| | 0.1 | 0.3015 | 0.2504 | 0.2392 | 0.2980 | 0.2509 | **0.4726** |
| | 0.2 | 0.3572 | 0.3587 | 0.2794 | 0.3322 | 0.2784 | **0.4684** |

Table 15: Robustness to hyperparameters for *Logic Conflict*. The results are averaged across 5 random seeds in classical control (Pendulum).

| [Logic Conflict ↓]
Network Architecture | | FCs | GRU | LSTM | ODE-RNN | CfC | NCP |
|---|---|---|---|---|---|---|---|
| Cost Complexity Pruning | 0.001 | 0.2451 | 0.3348 | 0.5240 | 0.2641 | **0.2048** | 0.3159 |
| | 0.003 | 0.2104 | 0.2832 | 0.5072 | 0.2506 | **0.1556** | 0.2026 |
| | 0.01 | 0.1766 | 0.1877 | 0.4325 | 0.1401 | **0.1121** | 0.2924 |
| Minimal Leaf Sample Ratio | 0.01 | 0.2672 | 0.4298 | 0.6791 | 0.3575 | **0.2654** | 0.2607 |
| | 0.1 | 0.2104 | 0.2832 | 0.5072 | 0.2506 | **0.1556** | 0.2026 |
| | 0.2 | 0.1796 | 0.1664 | 0.3842 | 0.2001 | **0.1089** | 0.1111 |

Table 16: Removing a single neuron based on explanation.

| Remove Neuron | 0 | 1 | 2 | 3 | 4 | 5 | 6 | 7 |
|---|---|---|---|---|---|---|---|---|
| Performance ↑ | 0.24 | 0.07 | 0.09 | 1.00 | 0.969 | 0.29 | 0.03 | 1.00 |

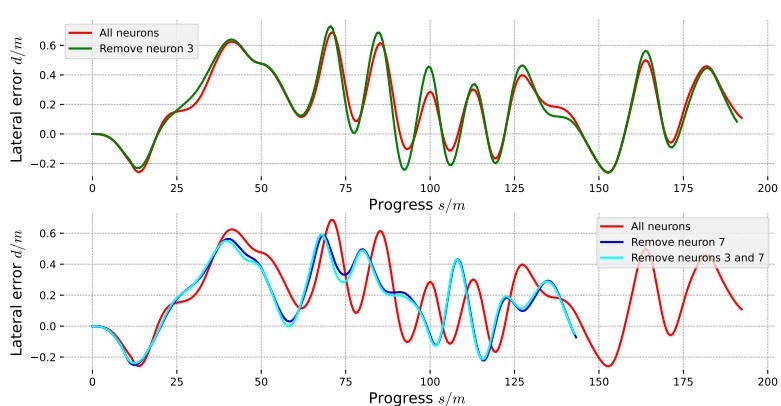

Figure 5: Driving profile when removing neurons according to decision tree interpretation.

this technique (e.g., 6B corresponds to CMD03 ¡= -0.914) to retrieve all time steps that fall into the branch 6A (early touchdown) or 6B (forward flipping) and compute average reward of the next few time steps. We also compute the non-branch results to serve as a reference, i.e., comparison between branch X and not branch X. For early touchdown (6A), we can observe from the quantitative results that the reward drops immediately when such a branch is "activated"; this makes sense as early touchdown brakes the robot right away, leading to smaller distance increments. For forward flipping (6B), we observe a higher reward at the closer time steps, which then quickly falls off to relatively much lower value; this is also reasonable as flipping motion carries the robot body forward a lot in the beginning stage yet leaves the robot body in a very bad pose to move forward afterward.

Table 17: Quantitative analysis on failure mode of locomotion.

| Step Run Reward At Time | t+1 | t+2 | t+3 | t+4 |
|---|---|---|---|---|
| Branch 6A (early touchdown) | 5.405 | 5.414 | 6.048 | 6.730 |
| Not Branch 6A | 6.235 | 6.233 | 6.031 | 5.813 |
| Branch 6B (forward flipping) | 7.145 | 6.431 | 5.392 | 4.998 |
| Not Branch 6B | 5.360 | 5.794 | 6.425 | 6.664 |

Table 18: Extension to larger models with Decision Transformer [60] as an example.

| Method | Variance | MI-Gap | Modularity |
|---|---|---|---|
| Decision Transformer | $0.007^{.003}$ | $0.167^{.088}$ | $0.981^{.017}$ |

## J  Potential Extension to Larger Models

While our work focuses only on compact neural networks, our method can be extended to larger-scale models by selecting a "bottleneck" layer and extract interpretation from neurons in that layer. Take transformers as examples. To start with, at a high level, transformer-based policies are constructed with an encoder-decoder architecture [59, 60] with inputs/outputs either being tokenized or kept to be directly mapped from continuous values to embeddings. The natural selection of the bottleneck is then the last hidden state of the encoder, which is a common way to do feature extraction from large transformer-based models [61, 62]. Overall, we believe there are promising extensions of our work toward larger-scale models.

Furthermore, in Table 18, we conduct an experiment using Decision Transformer [60] to demonstrate the future potential of our work. We apply our method to a pre-trained checkpoint that can achieve $\sim$ 10,000 reward in Gym Half-Cheetah. Given Decision Transformer adopts an encoder-decoder transformer-based architecture (similar to most language models), we extract the latest time step of the "last hidden state" of the encoder (following the terminology of Huggingface-Transformers here); specifically, last hidden state refers to the last layer of the stacked attention blocks and the latest time step refers to the last time step of the input sequence, e.g., in natural language, the "robot" in the sentence "I like robot". As discussed earlier, such technique is commonly adopted to extract features from large-scale transformer-based models. So far, in Decision Transformer, we get a (3, 128) feature, where 3 corresponds to return, action, and state prediction respectively. We take the dimension that is used to predict action, eventually leading to a 128-dimensional feature vector. We apply our method on this 128-dimensional feature vector and report the metrics.

Interestingly, Decision Transformer gives very good performance in neuron response variance and modularity, and slightly below average performance in MIG. Besides, we also show some samples of extracted logic programs. (Please refer to Section E.2 paragraph "Interpreter details" for detailed description of each symbol)

- $\dot{h}_R <= \text{-1.959} \wedge \theta_{S,B} <= \text{-0.256}$
- $\dot{\theta}_{S,B} <= \text{-2.139} \wedge \theta_{T,T} > \text{-0.995}$
- $\dot{\theta}_{S,B} <= \text{-0.945} \wedge \dot{h}_R <= 0.109$

Furthermore, as there are 128 neurons interpreted, which leads to a much larger set of possible decision paths and hence larger-sized logic programs, it drastically increases the cognitive load of humans interpreting the programs. To remedy this issue, we try to extract the decision path set from a smaller number of neurons by performing dimension reduction and only considering the principal components. Such an approximation can be surprisingly effective in practice as from the perspective of representation learning, the feature may exhibit a highly-structured distribution in the 128-dimensional space. We can empirically verify this by checking the explained variance (E.V.) of the principal components (P.C.) as shown in Table 19. We can see that the 10 principal components can achieve over 80% of explained variance. Note that the dimension reduction is only performed

Table 19: Explained variance of the dimension-reduced space of Decision Transformer's features.

| Principle Components | 1 | 2 | 3 | 4 | 5 | 6 | 7 | 8 | 9 | 10 |
|---|---|---|---|---|---|---|---|---|---|---|
| Explained Variance (E.V.) | 0.247 | 0.179 | 0.133 | 0.109 | 0.044 | 0.028 | 0.025 | 0.022 | 0.017 | 0.013 |
| Accumulated E.V. | 0.247 | 0.426 | 0.559 | 0.669 | 0.713 | 0.742 | 0.767 | 0.789 | 0.807 | 0.821 |

Table 20: Results for the human study.

| Method | Accuracy | Subjective Satisfaction |
|---|---|---|
| Non-NCP | $0.603^{.053}$ | $0.814^{.021}$ |
| NCP | $\mathbf{0.648}^{.065}$ | $\mathbf{0.971}^{.078}$ |

at constructing the decision path set for factors of variation and we are still interpreting all 128 neurons. We will then get much smaller-sized logic programs after performing the logic reduction step (as discussed in the Section D paragraph "Cross-neuron Logic Conflict"). Immediate research questions then arise here like *does this dimension reduction step still produce the factors of variation with similar amounts of information to the original ones?*, or *how sensitive is it to the methods and hyperparameters that extract the factors of variation?*, etc. These studies are extremely interesting yet go beyond this work and we leave these to future exploration.

While this additional study only provides a minimal experiment and analysis, we believe it demonstrates the potential of extending the proposed concept to larger-scale models and we will keep on exploring along this research direction in the future.

## K   Minimal Human Study as Validation

We design a questionnaire adapted from [41] to measure accuracy, response time, and subjective satisfaction. An example is shown in Figure 6. We show the human subject the observation of the policy (the image below what the robot sees) and the logic programs extracted from neuron responses (the text below In the mind of the robot), and ask the subject to guess what the robot will do next (the two different angles of the steering wheels). One of the options corresponds to the actual control predicted by the policy and the other is randomly sampled with the opposite sign from the actual control to avoid ambiguity. The user can choose between the two angles or non-selected (i.e., I don't know or I am not sure). Also, we put a checkbox below if the user thinks the logic program is not helping to measure subjective satisfaction. We record the answers of the subjects along with their response time.

The questionnaire consists of 144 questions, which takes roughly 20 minutes to finish. We sample 62 questions based on NCP's results and 62 questions based on non-NCP's since this specific architecture has the most distinction from the others across all tasks, thus potentially easier for minimal human experiment. We collected the results from 10 subjects, shown in the below table with superscript as standard deviation. Both accuracy and subjective satisfaction are between 0 and 1, where 0 is the worst and 1 is the best. The subjective satisfaction is the rate of the subject not checking the checkbox that indicates the logic program is not helping. The results are shown in Table 20. First, we can see that all accuracy are larger than 0.5 (random guess), which indicates that the logic programs are indeed useful for human users to understand the decision making of robots. Besides, we see positive correlation between subjective satisfaction and accuracy, which means when the human users think the logic programs are useful, they are indeed useful.

Note that, however, these results should be only viewed as a minimal experiment that augments the evaluation and analysis from [41, 42], which performs detailed user study on the human interpretability of the decision set (conceptually the same as logic programs used in our work). The

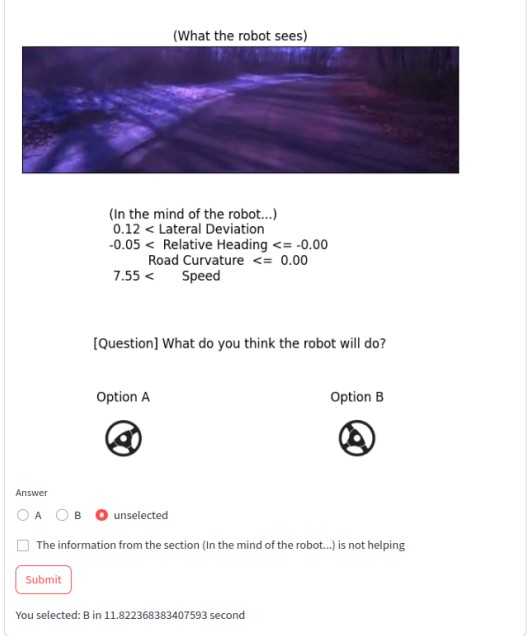

Figure 6: The demo of the user study questionnaire (adapted from [41]). We show the human subject the observation of the policy (the image below *what the robot sees*) and the logic programs extracted from neuron responses (the text below *In the mind of the robot*), and ask the subject to guess what the robot will do next (the two different angles of the steering wheels). One of the option corresponds to the actual control predicted by the policy and the other is randomly sampled with the opposite sign from the actual control to avoid ambiguity. The user can choose between the two angles or *unselected* (i.e., I don't know or I am not sure). Also, we put a checkbox below if the user thinks the logic program is not helping to measure subjective satisfaction. We record the answers of the subjects along with their response time.

more thorough and rigorous study with human subjects on interpretability in robot learning should be further explored in the future research.

## L    Logic Program from Decision Trees

Here we show the corresponding logic program of the finite set of decision path $\{r(\mathcal{P}_k^i)\}_{k=1}^{K^i}$ for every interpreted neuron in all network architectures. The symbols used in the logic program follow the state grounding definition in Section E. We also briefly summarize the size of associated decision trees by computing the number of decision rules for each model (before logic program reduction and conflict checking).

In classical control (Pendulum), the extracted logic program are shown in Table 21 (FC; of size 39), Table 22 (GRU; of size 54), Table 23 (LSTM; of size 43), Table 24 (ODE-RNN; of size 40), Table 25 (CfC; of size 20), Table 26 (NCP; of size 26).

In locomotion (HalfCheetah), the extracted logic program are shown in Table 27 (FC; of size 171), Table 28 (GRU; of size 158), Table 29 (LSTM; of size 148), Table 30 (ODE-RNN; of size 156), Table 31 (CfC; of size 149), Table 32 (NCP; of size 81).

In end-to-end visual servoing (Image-based Driving), the extracted logic program are shown in Table 33 (FC; of size 92), Table 34 (GRU; of size 60), Table 35 (LSTM; of size 70), Table 36 (ODE-RNN; of size 94), Table 37 (CfC; of size 107), Table 38 (NCP; of size 66).

In a logic program, "conflict" indicates there are conflict between predicates within the logic program as elaborated in Section 3.2.

| Model | Neuron | Logic Program |
|---|---|---|
| FC | 0 | 0: $(\dot{\theta} <= 0.69) \wedge (\theta <= -2.18)$
1: $(\dot{\theta} > 0.69) \wedge (\theta <= -2.18)$
2: (conflict)
3: $(\theta <= 2.41) \wedge (\theta > -2.18)$
4: $(\theta > 2.41)$ |
| | 1 | 0: $(\dot{\theta} <= -1.16) \wedge (\theta <= -0.34)$
1: $(\dot{\theta} <= 1.73) \wedge (\dot{\theta} > -1.16) \wedge (\theta <= -0.34)$
2: $(\dot{\theta} > 1.73) \wedge (\theta <= -0.34)$
3: $(\theta <= 2.03) \wedge (\theta > -0.34)$
4: $(\theta <= 2.62) \wedge (\theta > 2.03)$
5: $(\theta > 2.62)$ |
| | 2 | 0: $(\dot{\theta} <= -1.54) \wedge (\theta <= -1.68)$
1: $(\dot{\theta} <= 1.47) \wedge (\dot{\theta} > -1.54) \wedge (\theta <= -1.68)$
2: $(\dot{\theta} > 1.47) \wedge (\theta <= -1.68)$
3: (conflict)
4: $(\theta <= 2.48) \wedge (\theta > -1.68)$
5: $(\theta > 2.48)$ |
| | 3 | 0: $(\theta <= -2.76)$
1: (conflict)
2: $(\theta <= 0.05) \wedge (\theta > -2.76)$
3: $(\theta > 0.05)$ |

Table 21: Logic program of FC in classical control (Pendulum).

| Model | Neuron | Logic Program |
|---|---|---|
| GRU | 0 | 0: $(\theta <= -0.06)$
1: (conflict)
2: $(\dot{\theta} <= -0.30) \wedge (\theta > -0.06)$
3: $(\dot{\theta} <= 1.75) \wedge (\dot{\theta} > -0.30) \wedge (\theta > -0.06)$
4: $(\dot{\theta} > 1.75) \wedge (\theta > -0.06)$ |
| | 1 | 0: $(\dot{\theta} <= -2.30) \wedge (\theta <= -1.27)$
1: $(\dot{\theta} <= 1.83) \wedge (\dot{\theta} > -2.30) \wedge (\theta <= -1.27)$
2: $(\dot{\theta} <= -0.37) \wedge (\theta > -1.27)$
3: $(\dot{\theta} <= 1.83) \wedge (\dot{\theta} > -0.37) \wedge (\theta > -1.27)$
4: $(\dot{\theta} <= 3.10) \wedge (\dot{\theta} > 1.83)$
5: $(\dot{\theta} > 3.10)$ |
| | 2 | 0: $(\dot{\theta} <= -0.11) \wedge (\theta <= -0.05)$
1: $(\dot{\theta} <= -0.11) \wedge (\theta > -0.05)$
2: (conflict) $\wedge (\theta > -0.05)$
3: $(\dot{\theta} > -0.11) \wedge (\theta <= -2.09)$
4: $(\dot{\theta} > -0.11) \wedge (\theta <= 0.41) \wedge (\theta > -2.09)$
5: $(\dot{\theta} <= 1.61) \wedge (\dot{\theta} > -0.11) \wedge (\theta > 0.41)$
6: $(\dot{\theta} > 1.61) \wedge (\theta > 0.41)$ |
| | 3 | 0: $(\theta <= -2.61)$
1: $(\dot{\theta} <= 2.44) \wedge (\theta <= 0.21) \wedge (\theta > -2.61)$
2: $(\dot{\theta} > 2.44) \wedge (\theta <= 0.21) \wedge (\theta > -2.61)$
3: $(\dot{\theta} <= -1.76) \wedge (\theta > 0.21)$
4: $(\dot{\theta} <= 0.39) \wedge (\dot{\theta} > -1.76) \wedge (\theta > 0.21)$
5: $(\dot{\theta} > 0.39) \wedge (\theta > 0.21)$ |

Table 22: Logic program of GRU in classical control (Pendulum).

| Model | Neuron | Logic Program |
|---|---|---|
| LSTM | 0 | 0: $(\theta <= -2.16)$
1: (conflict)
2: $(\theta <= 0.01) \wedge (\theta > -2.16)$
3: $(\dot{\theta} <= 0.48) \wedge (\theta > 0.01)$
4: $(\dot{\theta} <= 3.00) \wedge (\dot{\theta} > 0.48) \wedge (\theta > 0.01)$
5: $(\dot{\theta} > 3.00) \wedge (\theta > 0.01)$ |
| | 1 | 0: $(\dot{\theta} <= -2.57)$
1: (conflict)
2: $(\dot{\theta} > -2.57) \wedge (\theta <= 0.35)$
3: $(\dot{\theta} > -2.57) \wedge (\theta <= 2.03) \wedge (\theta > 0.35)$
4: $(\dot{\theta} > -2.57) \wedge (\theta > 2.03)$ |
| | 2 | 0: $(\dot{\theta} <= 4.79) \wedge (\theta <= -2.31)$
1: $(\dot{\theta} <= 4.79) \wedge (\theta <= 1.72) \wedge (\theta > -2.31)$
2: $(\dot{\theta} <= -3.13) \wedge (\theta > 1.72)$
3: $(\dot{\theta} <= 4.79) \wedge (\dot{\theta} > -3.13) \wedge (\theta > 1.72)$
4: $(\dot{\theta} > 4.79)$ |
| | 3 | 0: $(\theta <= -2.32)$
1: $(\theta <= 0.93) \wedge (\theta > -2.32)$
2: $(\dot{\theta} <= -3.98) \wedge (\theta > 0.93)$
3: $(\dot{\theta} > -3.98) \wedge (\theta <= 2.04) \wedge (\theta > 0.93)$
4: $(\dot{\theta} > -3.98) \wedge (\theta > 2.04)$ |

Table 23: Logic program of LSTM in classical control (Pendulum).

| Model | Neuron | Logic Program |
|---|---|---|
| ODE-RNN | 0 | 0: $(\dot{\theta} <= -1.48) \wedge (\theta <= -0.02)$
1: $(\dot{\theta} <= -1.48) \wedge (\theta > -0.02)$
2: $(\dot{\theta} > -1.48)$ |
| | 1 | 0: $(\dot{\theta} <= -0.08) \wedge (\theta <= -1.45)$
1: $(\dot{\theta} > -0.08) \wedge (\theta <= -1.45)$
2: $(\theta <= 2.05) \wedge (\theta > -1.45)$
3: $(\theta <= 2.50) \wedge (\theta > 2.05)$
4: $(\dot{\theta} <= -0.40) \wedge (\theta > 2.50)$
5: $(\dot{\theta} <= 0.03) \wedge (\dot{\theta} > -0.40) \wedge (\theta > 2.50)$
6: $(\dot{\theta} > 0.03) \wedge (\theta > 2.50)$ |
| | 2 | 0: $(\dot{\theta} <= -0.56)$
1: $(\dot{\theta} > -0.56) \wedge (\theta <= -2.16)$
2: $(\dot{\theta} > -0.56) \wedge (\theta <= 2.44) \wedge (\theta > -2.16)$
3: (conflict) $\wedge (\theta > 2.44)$
4: $(\dot{\theta} > -0.56) \wedge (\theta > 2.44)$ |
| | 3 | 0: $(\theta <= -2.18)$
1: $(\theta <= 0.04) \wedge (\theta > -2.18)$
2: $(\theta <= 2.65) \wedge (\theta > 0.04)$
3: $(\dot{\theta} <= -0.21) \wedge (\theta > 2.65)$
4: $(\dot{\theta} > -0.21) \wedge (\theta > 2.65)$ |

Table 24: Logic program of ODE-RNN in classical control (Pendulum).

| Model | Neuron | Logic Program |
|-------|--------|---------------|
| CfC | 0 | 0: ($\theta <= -0.03$)
1: (conflict)
2: (conflict)
3: ($\theta > -0.03$) |
| | 1 | 0: ($\theta <= -0.05$)
1: (conflict)
2: (conflict)
3: ($\theta > -0.05$) |
| | 2 | 0: ($\theta <= -2.02$)
1: ($\theta > -2.02$) |
| | 3 | 0: ($\theta <= -0.12$)
1: ($\theta <= 0.24$) $\land$ ($\theta > -0.12$)
2: ($\theta <= 2.14$) $\land$ ($\theta > 0.24$)
3: ($\theta > 2.14$) |

Table 25: Logic program of CfC in classical control (Pendulum).

| Model | Neuron | Logic Program |
|-------|--------|---------------|
| NCP | 0 | 0: ($\dot{\theta} <= 0.33$)
1: ($\dot{\theta} > 0.33$) |
| | 1 | 0: ($\theta <= -0.07$)
1: (conflict)
2: ($\theta <= 0.27$) $\land$ ($\theta > -0.07$)
3: ($\theta > 0.27$) |
| | 2 | 0: ($\dot{\theta} <= 4.80$) $\land$ ($\theta <= -1.27$)
1: ($\dot{\theta} <= 4.80$) $\land$ ($\theta <= 1.66$) $\land$ ($\theta > -1.27$)
2: ($\dot{\theta} <= 4.80$) $\land$ ($\theta > 1.66$)
3: ($\dot{\theta} > 4.80$) |
| | 3 | 0: ($\dot{\theta} <= -0.33$)
1: ($\dot{\theta} <= 0.44$) $\land$ ($\dot{\theta} > -0.33$)
2: ($\dot{\theta} > 0.44$) $\land$ ($\theta <= -1.31$)
3: ($\dot{\theta} > 0.44$) $\land$ ($\theta <= 1.44$) $\land$ ($\theta > -1.31$)
4: ($\dot{\theta} > 0.44$) $\land$ ($\theta > 1.44$) |

Table 26: Logic program of NCP in classical control (Pendulum).

| Model | Neuron | Logic Program |
|-------|--------|---------------|
| FC | 0 | 0: $(\dot{\theta}_R <= -0.22) \wedge (\dot{\theta}_{T,F} <= -3.26)$ 
 1: $(\dot{\theta}_R > -0.22) \wedge (\dot{\theta}_{T,F} <= -3.26)$ 
 2: $(\dot{\theta}_{T,F} <= 6.46) \wedge (\dot{\theta}_{T,F} > -3.26) \wedge (\theta_{F,F} <= -0.50)$ 
 3: $(\dot{\theta}_{T,F} <= 6.46) \wedge (\dot{\theta}_{T,F} > -3.26) \wedge (\theta_{F,F} > -0.50)$ 
 4: $(\dot{\theta}_{T,F} > 6.46) \wedge (h_R <= 0.05)$ 
 5: $(\dot{\theta}_{T,F} > 6.46) \wedge (h_R > 0.05)$ |
| | 1 | 0: $(\theta_{F,B} <= -0.05) \wedge (\theta_{S,F} <= 0.61) \wedge (\theta_{T,B} <= 0.05)$ 
 1: $(\theta_{F,B} <= -0.05) \wedge (\theta_{S,F} > 0.61) \wedge (\theta_{T,B} <= 0.05)$ 
 2: $(\dot{\theta}_{T,F} <= -11.24) \wedge (\theta_{F,B} > -0.05) \wedge (\theta_{T,B} <= 0.05)$ 
 3: $(\dot{\theta}_{T,F} > -11.24) \wedge (\theta_{F,B} > -0.05) \wedge (\theta_{T,B} <= 0.05)$ 
 4: $(\theta_{T,B} > 0.05) \wedge (\theta_{T,F} <= 0.40)$ 
 5: $(\theta_{T,B} > 0.05) \wedge (\theta_{T,F} <= 0.62) \wedge (\theta_{T,F} > 0.40)$ 
 6: $(\dot{\theta}_{T,B} <= 2.08) \wedge (\theta_{T,B} > 0.05) \wedge (\theta_{T,F} > 0.62)$ 
 7: $(\dot{\theta}_{T,B} > 2.08) \wedge (\theta_{T,B} > 0.05) \wedge (\theta_{T,F} > 0.62)$ |
| | 2 | 0: $(\dot{\theta}_{F,F} <= -12.45) \wedge (\theta_{F,B} <= 0.06)$ 
 1: $(\dot{\theta}_{F,F} > -12.45) \wedge (\theta_{F,B} <= 0.06) \wedge (\theta_{S,F} <= 0.65)$ 
 2: $(\dot{\theta}_{F,F} > -12.45) \wedge (\theta_{F,B} <= 0.06) \wedge (\theta_{S,F} > 0.65)$ 
 3: $(\dot{\theta}_{T,B} <= 6.33) \wedge (\theta_{F,B} > 0.06) \wedge (\theta_{T,F} <= 0.33)$ 
 4: $(\dot{\theta}_{T,B} <= 6.33) \wedge (\theta_{F,B} > 0.06) \wedge (\theta_{T,F} > 0.33)$ 
 5: $(\dot{\theta}_{T,B} > 6.33) \wedge (\theta_{F,B} > 0.06)$ |
| | 3 | 0: $(\theta_{F,B} <= 0.38) \wedge (\theta_{S,F} <= 0.17) \wedge (\theta_{T,F} <= 0.58)$ 
 1: $(\theta_{F,B} <= 0.38) \wedge (\theta_{S,F} <= 0.17) \wedge (\theta_{T,F} > 0.58)$ 
 2: $(\theta_{F,B} <= 0.38) \wedge (\theta_{S,F} > 0.17) \wedge (\theta_{T,B} <= 0.07)$ 
 3: $(\theta_{F,B} <= 0.38) \wedge (\theta_{S,F} > 0.17) \wedge (\theta_{T,B} > 0.07)$ 
 4: $(\theta_{F,B} > 0.38)$ |
| | 4 | 0: $(\dot{\theta}_{S,B} <= 1.79) \wedge (\theta_R <= 0.19) \wedge (\theta_{T,B} <= 0.06)$ 
 1: $(\dot{\theta}_{S,B} > 1.79) \wedge (\theta_R <= 0.19) \wedge (\theta_{T,B} <= 0.06)$ 
 2: $(\dot{\theta}_{F,F} <= 9.07) \wedge (\theta_R > 0.19) \wedge (\theta_{T,B} <= 0.06)$ 
 3: $(\dot{\theta}_{F,F} > 9.07) \wedge (\theta_R > 0.19) \wedge (\theta_{T,B} <= 0.06)$ 
 4: $(\theta_R <= -0.03) \wedge (\theta_{T,B} > 0.06)$ 
 5: $(\theta_R > -0.03) \wedge (\theta_{F,F} <= -0.49) \wedge (\theta_{T,B} > 0.06)$ 
 6: $(\theta_R > -0.03) \wedge (\theta_{F,F} > -0.49) \wedge (\theta_{T,B} > 0.06)$ |
| | 5 | 0: $(\dot{\theta}_{T,B} <= 1.10) \wedge (\dot{\theta}_{T,F} <= -6.72) \wedge (\theta_{T,F} <= 0.67)$ 
 1: $(\dot{\theta}_{T,B} <= 1.10) \wedge (\dot{\theta}_{T,F} > -6.72) \wedge (\theta_{T,F} <= 0.67)$ 
 2: $(\dot{\theta}_{T,B} <= 1.10) \wedge (\theta_{T,F} > 0.67)$ 
 3: $(\dot{\theta}_{T,B} > 1.10) \wedge (\dot{h}_R <= -0.33) \wedge (\theta_R <= 0.29)$ 
 4: $(\dot{\theta}_{T,B} > 1.10) \wedge (\dot{h}_R > -0.33) \wedge (\theta_R <= 0.29)$ 
 5: $(\dot{\theta}_{T,B} > 1.10) \wedge (\dot{h}_R <= -0.48) \wedge (\theta_R > 0.29)$ 
 6: $(\dot{\theta}_{T,B} > 1.10) \wedge (\dot{h}_R > -0.48) \wedge (\theta_R > 0.29)$ |
| | 6 | 0: $(\dot{\theta}_R <= -0.83) \wedge (\dot{\theta}_{F,B} <= 4.65) \wedge (\dot{\theta}_{F,F} <= -0.07)$ 
 1: $(\dot{\theta}_R <= -0.83) \wedge (\dot{\theta}_{F,B} > 4.65) \wedge (\dot{\theta}_{F,F} <= -0.07)$ 
 2: $(\dot{\theta}_R > -0.83) \wedge (\dot{\theta}_{F,F} <= -0.07) \wedge (\theta_{S,B} <= -0.33)$ 
 3: $(\dot{\theta}_R > -0.83) \wedge (\dot{\theta}_{F,F} <= -0.07) \wedge (\theta_{S,B} > -0.33)$ 
 4: $(\dot{\theta}_{F,F} > -0.07) \wedge (\theta_R <= 0.52)$ 
 5: $(\dot{\theta}_{F,F} > -0.07) \wedge (\theta_R > 0.52)$ |
| | 7 | 0: $(\dot{\theta}_{T,F} <= -0.36) \wedge (\theta_R <= 0.33) \wedge (\theta_{S,F} <= 0.56)$ 
 1: $(\dot{\theta}_{T,F} <= -0.36) \wedge (\theta_R > 0.33) \wedge (\theta_{S,F} <= 0.56)$ 
 2: $(\dot{\theta}_{T,F} <= -0.36) \wedge (\theta_{S,F} > 0.56)$ 
 3: $(\dot{\theta}_{T,F} <= 6.91) \wedge (\dot{\theta}_{T,F} > -0.36) \wedge (\theta_{S,B} <= 0.09)$ 
 4: $(\dot{\theta}_{T,F} > 6.91) \wedge (\theta_{S,B} <= 0.09)$ 
 5: $(\dot{\theta}_{T,F} > -0.36) \wedge (\theta_{S,B} > 0.09)$ |
| | 8 | 0: $(\dot{\theta}_R <= -0.10) \wedge (\dot{\theta}_{T,B} <= -3.31)$ 
 1: $(\dot{\theta}_R > -0.10) \wedge (\dot{\theta}_{T,B} <= -3.31)$ 
 2: $(\dot{\theta}_{T,B} <= 0.97) \wedge (\dot{\theta}_{T,B} > -3.31)$ 
 3: $(\dot{\theta}_{T,B} > 0.97) \wedge (\theta_{S,B} <= -0.35)$ 
 4: $(\dot{\theta}_{T,B} > 0.97) \wedge (\theta_R <= 0.20) \wedge (\theta_{S,B} > -0.35)$ 
 5: $(\dot{\theta}_{T,B} > 0.97) \wedge (\theta_R > 0.20) \wedge (\theta_{S,B} > -0.35)$ |
| | 9 | 0: $(\dot{\theta}_{F,B} <= 0.63) \wedge (\dot{\theta}_{S,F} <= 3.07)$ 
 1: $(\dot{\theta}_{F,B} <= 0.63) \wedge (\dot{\theta}_{S,F} > 3.07) \wedge (\theta_R <= 0.46)$ 
 2: $(\dot{\theta}_{F,B} <= 0.63) \wedge (\dot{\theta}_{S,F} > 3.07) \wedge (\theta_R > 0.46)$ 
 3: $(\dot{\theta}_{F,B} > 0.63) \wedge (\dot{\theta}_{T,F} <= 5.96) \wedge (h_R <= 0.03)$ 
 4: $(\dot{\theta}_{F,B} > 0.63) \wedge (\dot{\theta}_{T,F} <= 5.96) \wedge (h_R > 0.03)$ 
 5: $(\dot{\theta}_{F,B} > 0.63) \wedge (\dot{\theta}_{T,F} > 5.96) \wedge (\theta_{T,F} <= 0.38)$ 
 6: $(\dot{\theta}_{F,B} > 0.63) \wedge (\dot{\theta}_{T,F} > 5.96) \wedge (\theta_{T,F} > 0.38)$ |

Table 27: Logic program of FC in locomotion (HalfCheetah).

| Model | Neuron | Logic Program |
|---|---|---|
| | 0 | 0: $(\dot{\theta}_R <= 1.54) \wedge (\dot{\theta}_{T,B} <= -2.33) \wedge (\theta_R <= 0.52)$
1: $(\dot{\theta}_R > 1.54) \wedge (\dot{\theta}_{T,B} <= -2.33) \wedge (\theta_R <= 0.52)$
2: $(\dot{\theta}_{T,B} > -2.33) \wedge (\theta_R <= 0.11)$
3: $(\dot{\theta}_{T,B} > -2.33) \wedge (\theta_R <= 0.52) \wedge (\theta_R > 0.11)$
4: $(\dot{\theta}_{T,F} <= -7.48) \wedge (\theta_R > 0.52)$
5: $(\dot{\theta}_{T,F} > -7.48) \wedge (\theta_R <= 0.97) \wedge (\theta_R > 0.52)$
6: $(\dot{\theta}_{T,F} > -7.48) \wedge (\theta_R > 0.97)$ |
| | 1 | 0: $(\dot{\theta}_{S,B} <= 10.33) \wedge (\theta_R <= 0.50) \wedge (\theta_{F,B} <= -0.41)$
1: $(\dot{\theta}_{S,B} <= 10.33) \wedge (\theta_R <= 0.50) \wedge (\theta_{F,B} > -0.41)$
2: $(\dot{\theta}_{S,B} <= 10.33) \wedge (\dot{h}_R <= -0.07) \wedge (\theta_R > 0.50)$
3: $(\dot{\theta}_{S,B} <= 10.33) \wedge (\dot{h}_R > -0.07) \wedge (\theta_R > 0.50)$
4: $(\dot{\theta}_{S,B} > 10.33)$ |
| | 2 | 0: $(\dot{\theta}_{T,B} <= 3.33) \wedge (\theta_R <= 0.12)$
1: $(\dot{\theta}_{T,B} > 3.33) \wedge (\theta_R <= 0.12)$
2: $(\dot{\theta}_{T,B} <= 6.59) \wedge (\theta_R > 0.12) \wedge (\theta_{T,F} <= 0.70)$
3: $(\dot{\theta}_{T,B} <= 6.59) \wedge (\theta_R > 0.12) \wedge (\theta_{T,F} > 0.70)$
4: $(\dot{\theta}_{T,B} > 6.59) \wedge (\theta_R <= 0.54) \wedge (\theta_R > 0.12)$
5: $(\dot{\theta}_{T,B} > 6.59) \wedge (\theta_R > 0.54)$ |
| | 3 | 0: $(\dot{\theta}_R <= -0.78) \wedge (\theta_{T,F} <= 0.17)$
1: $(\dot{\theta}_R > -0.78) \wedge (\theta_R <= 0.68) \wedge (\theta_{T,F} <= 0.17)$
2: $(\dot{\theta}_R > -0.78) \wedge (\theta_R > 0.68) \wedge (\theta_{T,F} <= 0.17)$
3: $(\dot{h}_R <= 0.64) \wedge (\theta_{T,B} <= -0.14) \wedge (\theta_{T,F} > 0.17)$
4: $(\dot{h}_R <= 0.64) \wedge (\theta_{T,B} > -0.14) \wedge (\theta_{T,F} > 0.17)$
5: $(\dot{h}_R > 0.64) \wedge (\theta_{T,F} > 0.17)$ |
| GRU | 4 | 0: $(\dot{\theta}_{S,B} <= 1.92) \wedge (\theta_{S,F} <= 0.02)$
1: $(\dot{\theta}_{S,B} > 1.92) \wedge (\theta_{S,F} <= 0.02)$
2: $(\dot{\theta}_{S,B} <= 6.10) \wedge (\dot{\theta}_{S,F} <= 7.21) \wedge (\theta_{S,F} > 0.02)$
3: $(\dot{\theta}_{S,B} <= 6.10) \wedge (\dot{\theta}_{S,F} > 7.21) \wedge (\theta_{S,F} > 0.02)$
4: $(\dot{\theta}_{S,B} > 6.10) \wedge (\theta_{S,F} > 0.02)$ |
| | 5 | 0: $(\dot{\theta}_{T,B} <= 2.59) \wedge (\theta_{F,B} <= 0.10) \wedge (\theta_{T,B} <= -0.16)$
1: $(\dot{\theta}_{T,B} <= 2.59) \wedge (\theta_{F,B} <= 0.10) \wedge (\theta_{T,B} > -0.16)$
2: $(\dot{\theta}_{T,B} > 2.59) \wedge (\theta_{F,B} <= 0.10)$
3: $(\dot{\theta}_{T,B} <= 1.45) \wedge (\dot{\theta}_{T,F} <= 6.43) \wedge (\theta_{F,B} > 0.10)$
4: $(\dot{\theta}_{T,B} > 1.45) \wedge (\dot{\theta}_{T,F} <= 6.43) \wedge (\theta_{F,B} > 0.10)$
5: (conflict) $\wedge (\theta_{F,B} > 0.10)$
6: $(\dot{\theta}_{T,F} > 6.43) \wedge (\theta_{F,B} > 0.10)$ |
| | 6 | 0: $(\theta_R <= 0.17) \wedge (\theta_{F,B} <= -0.12)$
1: $(\theta_R <= 0.62) \wedge (\theta_R > 0.17) \wedge (\theta_{F,B} <= -0.12)$
2: $(\dot{\theta}_{F,B} <= 6.66) \wedge (\theta_R <= 0.62) \wedge (\theta_{F,B} > -0.12)$
3: $(\dot{\theta}_{F,B} > 6.66) \wedge (\theta_R <= 0.62) \wedge (\theta_{F,B} > -0.12)$
4: $(\dot{\theta}_{T,B} <= -5.47) \wedge (\theta_R > 0.62)$
5: $(\dot{\theta}_{T,B} > -5.47) \wedge (\theta_R <= 0.89) \wedge (\theta_R > 0.62)$
6: $(\dot{\theta}_{T,B} > -5.47) \wedge (\theta_R > 0.89)$ |
| | 7 | 0: $(\dot{h}_R <= -0.27) \wedge (\theta_{T,F} <= 0.20)$
1: $(\dot{\theta}_{T,B} <= -0.25) \wedge (\dot{h}_R > -0.27) \wedge (\theta_{T,F} <= 0.20)$
2: $(\dot{\theta}_{T,B} > -0.25) \wedge (\dot{h}_R > -0.27) \wedge (\theta_{T,F} <= 0.20)$
3: $(\dot{h}_R <= 0.12) \wedge (\theta_R <= 0.77) \wedge (\theta_{T,F} > 0.20)$
4: $(\dot{h}_R <= 0.12) \wedge (\theta_R > 0.77) \wedge (\theta_{T,F} > 0.20)$
5: $(\dot{h}_R > 0.12) \wedge (\theta_{T,F} > 0.20)$ |
| | 8 | 0: $(\dot{h}_R <= 0.13) \wedge (\theta_{F,B} <= -0.16) \wedge (h_R <= 0.07)$
1: $(\dot{h}_R <= 0.13) \wedge (\theta_{F,B} > -0.16) \wedge (h_R <= 0.07)$
2: $(\dot{h}_R <= 0.13) \wedge (h_R > 0.07)$
3: $(\dot{h}_R <= 1.11) \wedge (\dot{h}_R > 0.13) \wedge (h_R <= 0.08)$
4: $(\dot{h}_R > 1.11) \wedge (h_R <= 0.08)$
5: $(\dot{h}_R > 0.13) \wedge (h_R > 0.08)$ |
| | 9 | 0: $(\dot{\theta}_{T,F} <= 0.51) \wedge (\theta_{S,F} <= -0.07)$
1: $(\dot{\theta}_{F,F} <= 3.81) \wedge (\dot{\theta}_{T,F} <= 0.51) \wedge (\theta_{S,F} > -0.07)$
2: $(\dot{\theta}_{F,F} > 3.81) \wedge (\dot{\theta}_{T,F} <= 0.51) \wedge (\theta_{S,F} > -0.07)$
3: $(\dot{\theta}_{T,F} > 0.51) \wedge (\theta_{F,B} <= 0.34)$
4: $(\dot{\theta}_R <= -1.78) \wedge (\dot{\theta}_{T,F} > 0.51) \wedge (\theta_{F,B} > 0.34)$
5: $(\dot{\theta}_R > -1.78) \wedge (\dot{\theta}_{T,F} > 0.51) \wedge (\theta_{F,B} > 0.34)$ |

Table 28: Logic program of GRU in locomotion (HalfCheetah).

| Model | Neuron | Logic Program |
|-------|--------|---------------|
| LSTM | 0 | 0: $(\dot\theta_R <= 2.01) \wedge (\theta_{S,B} <= 0.53) \wedge (\theta_{T,F} <= -0.28)$
1: $(\dot\theta_R <= 2.01) \wedge (\theta_{S,B} <= 0.53) \wedge (\theta_{T,F} > -0.28)$
2: $(\dot\theta_R > 2.01) \wedge (\theta_{S,B} <= 0.53)$
3: $(\theta_{S,B} > 0.53)$ |
| | 1 | 0: $(\theta_{F,F} <= 0.02) \wedge (\theta_{T,F} <= -0.35)$
1: $(\dot\theta_{F,F} <= 1.23) \wedge (\theta_{F,F} <= 0.02) \wedge (\theta_{T,F} > -0.35)$
2: $(\dot\theta_{F,F} > 1.23) \wedge (\theta_{F,F} <= 0.02) \wedge (\theta_{T,F} > -0.35)$
3: $(\theta_{F,F} > 0.02) \wedge (\theta_{T,F} <= -0.07)$
4: $(\dot\theta_{F,F} <= 0.95) \wedge (\theta_{F,F} > 0.02) \wedge (\theta_{T,F} > -0.07)$
5: $(\dot\theta_{F,F} > 0.95) \wedge (\theta_{F,F} > 0.02) \wedge (\theta_{T,F} > -0.07)$ |
| | 2 | 0: $(\theta_{F,F} <= -0.46) \wedge (h_R <= -0.07)$
1: $(\dot\theta_{F,B} <= -4.76) \wedge (\theta_{F,F} > -0.46) \wedge (h_R <= -0.07)$
2: $(\dot\theta_{F,B} > -4.76) \wedge (\theta_{F,F} > -0.46) \wedge (h_R <= -0.07)$
3: $(\theta_R <= 0.11) \wedge (h_R > -0.07)$
4: $(\theta_R > 0.11) \wedge (h_R > -0.07)$ |
| | 3 | 0: $(\dot\theta_{S,B} <= 0.14) \wedge (\theta_{T,F} <= -0.01) \wedge (h_R <= -0.11)$
1: $(\dot\theta_{S,B} <= 0.14) \wedge (\theta_{T,F} > -0.01) \wedge (h_R <= -0.11)$
2: $(\dot\theta_{S,B} > 0.14) \wedge (\theta_{S,B} <= 0.42) \wedge (h_R <= -0.11)$
3: $(\dot\theta_{S,B} > 0.14) \wedge (\theta_{S,B} > 0.42) \wedge (h_R <= -0.11)$
4: $(\theta_{T,B} <= 0.27) \wedge (h_R > -0.11)$
5: $(\theta_{T,B} <= 0.53) \wedge (\theta_{T,B} > 0.27) \wedge (h_R > -0.11)$
6: $(\theta_{T,B} > 0.53) \wedge (h_R > -0.11)$ |
| | 4 | 0: $(\theta_{F,F} <= -0.00)$
1: (conflict)
2: $(\theta_R <= 0.12) \wedge (\theta_{F,F} <= 0.39) \wedge (\theta_{F,F} > -0.00)$
3: $(\theta_R > 0.12) \wedge (\theta_{F,F} <= 0.39) \wedge (\theta_{F,F} > -0.00)$
4: $(\dot\theta_R <= -0.51) \wedge (\theta_{F,F} > 0.39)$
5: $(\dot\theta_R > -0.51) \wedge (\theta_{F,F} > 0.39)$ |
| | 5 | 0: $(\dot\theta_R <= 0.07) \wedge (\theta_{F,F} <= -0.05)$
1: $(\dot\theta_R <= 0.07) \wedge (\theta_{F,F} <= 0.35) \wedge (\theta_{F,F} > -0.05)$
2: $(\dot\theta_R > 0.07) \wedge (\theta_{F,F} <= 0.35) \wedge (h_R <= -0.18)$
3: $(\dot\theta_R > 0.07) \wedge (\theta_{F,F} <= 0.35) \wedge (h_R > -0.18)$
4: $(\dot\theta_{S,B} <= -2.15) \wedge (\theta_{F,F} > 0.35)$
5: $(\dot\theta_{S,B} > -2.15) \wedge (\theta_{F,F} > 0.35)$ |
| | 6 | 0: $(\dot\theta_{T,F} <= 1.81) \wedge (\dot h_R <= -1.08) \wedge (\theta_{T,F} <= -0.14)$
1: $(\dot\theta_{T,F} <= 1.81) \wedge (\dot h_R > -1.08) \wedge (\theta_{T,F} <= -0.14)$
2: $(\dot\theta_{T,F} > 1.81) \wedge (\theta_{T,F} <= -0.14)$
3: $(\dot h_R <= -0.47) \wedge (\theta_{T,F} > -0.14)$
4: $(\dot\theta_{T,B} <= -1.89) \wedge (\dot h_R > -0.47) \wedge (\theta_{T,F} > -0.14)$
5: $(\dot\theta_{T,B} > -1.89) \wedge (\dot h_R > -0.47) \wedge (\theta_{T,F} > -0.14)$ |
| | 7 | 0: $(\theta_{F,F} <= -0.07) \wedge (\theta_{T,F} <= 0.36)$
1: $(\theta_{F,F} <= -0.07) \wedge (\theta_{T,F} > 0.36)$
2: $(\dot x_R <= 4.14) \wedge (\theta_{F,F} > -0.07) \wedge (\theta_{T,B} <= 0.34)$
3: $(\dot x_R > 4.14) \wedge (\theta_{F,F} > -0.07) \wedge (\theta_{T,B} <= 0.34)$
4: $(\dot\theta_{T,F} <= -2.37) \wedge (\theta_{F,F} > -0.07) \wedge (\theta_{T,B} > 0.34)$
5: $(\dot\theta_{T,F} > -2.37) \wedge (\theta_{F,F} > -0.07) \wedge (\theta_{T,B} > 0.34)$ |
| | 8 | 0: $(\dot\theta_{F,B} <= 11.09) \wedge (\dot\theta_{F,F} <= 1.05) \wedge (\theta_{T,F} <= -0.46)$
1: $(\dot\theta_{F,B} <= 11.09) \wedge (\dot\theta_{F,F} <= 1.05) \wedge (\theta_{T,F} > -0.46)$
2: $(\dot\theta_{F,B} > 11.09) \wedge (\dot\theta_{F,F} <= 1.05)$
3: $(\dot\theta_{F,F} > 1.05) \wedge (\dot\theta_{T,B} <= -11.38)$
4: $(\dot\theta_{F,F} > 1.05) \wedge (\dot\theta_{T,B} > -11.38) \wedge (\theta_R <= 0.16)$
5: $(\dot\theta_{F,F} > 1.05) \wedge (\dot\theta_{T,B} > -11.38) \wedge (\theta_R > 0.16)$ |
| | 9 | 0: $(\theta_{F,F} <= -0.40) \wedge (\theta_{S,B} <= -0.33)$
1: $(\theta_{F,F} <= -0.40) \wedge (\theta_{S,B} > -0.33)$
2: $(\dot\theta_{T,F} <= -10.21) \wedge (\dot h_R <= -0.80) \wedge (\theta_{F,F} > -0.40)$
3: $(\dot\theta_{T,F} > -10.21) \wedge (\dot h_R <= -0.80) \wedge (\theta_{F,F} > -0.40)$
4: $(\dot h_R > -0.80) \wedge (\theta_{F,B} <= -0.14) \wedge (\theta_{F,F} > -0.40)$
5: $(\dot h_R > -0.80) \wedge (\theta_{F,B} > -0.14) \wedge (\theta_{F,F} > -0.40)$ |

Table 29: Logic program of LSTM in locomotion (HalfCheetah).

| Model | Neuron | Logic Program |
|-------|--------|---------------|
| ODE-RNN | 0 | 0: $(\dot{h}_R <= -0.27) \wedge (\theta_{S,B} <= -0.49)$
1: $(\dot{h}_R > -0.27) \wedge (\theta_{S,B} <= -0.49)$
2: $(\dot{h}_R <= -0.00) \wedge (\theta_{F,F} <= -0.22) \wedge (\theta_{S,B} > -0.49)$
3: $(\dot{h}_R <= -0.00) \wedge (\theta_{F,F} > -0.22) \wedge (\theta_{S,B} > -0.49)$
4: $(\dot{h}_R > -0.00) \wedge (\theta_{S,B} > -0.49) \wedge (\theta_{T,B} <= 0.22)$
5: $(\dot{h}_R > -0.00) \wedge (\theta_{S,B} > -0.49) \wedge (\theta_{T,B} > 0.22)$ |
| | 1 | 0: $(\dot{\theta}_{T,B} <= -2.91) \wedge (\theta_{F,B} <= -0.39)$
1: $(\dot{\theta}_{T,B} > -2.91) \wedge (\dot{h}_R <= -0.21) \wedge (\theta_{F,B} <= -0.39)$
2: $(\dot{\theta}_{T,B} > -2.91) \wedge (\dot{h}_R > -0.21) \wedge (\theta_{F,B} <= -0.39)$
3: $(\dot{\theta}_R <= -0.00) \wedge (\theta_{F,B} > -0.39)$
4: $(\dot{\theta}_R > -0.00) \wedge (\theta_R <= -0.04) \wedge (\theta_{F,B} > -0.39)$
5: $(\dot{\theta}_R > -0.00) \wedge (\theta_R > -0.04) \wedge (\theta_{F,B} > -0.39)$ |
| | 2 | 0: $(\theta_R <= 0.02) \wedge (\theta_{T,B} <= -0.16)$
1: $(\theta_R <= 0.02) \wedge (\theta_{T,B} > -0.16) \wedge (h_R <= 0.00)$
2: $(\theta_R <= 0.02) \wedge (\theta_{T,B} > -0.16) \wedge (h_R > 0.00)$
3: $(\theta_R > 0.02) \wedge (h_R <= -0.02)$
4: $(\theta_R > 0.02) \wedge \text{(conflict)}$
5: $(\theta_R > 0.02) \wedge (h_R > -0.02)$ |
| | 3 | 0: $(\dot{h}_R <= 0.55) \wedge (\theta_R <= 0.08) \wedge (\theta_{T,B} <= 0.42)$
1: $(\dot{h}_R > 0.55) \wedge (\theta_R <= 0.08) \wedge (\theta_{T,B} <= 0.42)$
2: $(\theta_R <= 0.08) \wedge (\theta_{S,B} <= 0.16) \wedge (\theta_{T,B} > 0.42)$
3: $(\theta_R <= 0.08) \wedge (\theta_{S,B} > 0.16) \wedge (\theta_{T,B} > 0.42)$
4: $(\theta_R > 0.08) \wedge (\theta_{T,B} <= 0.42) \wedge (h_R <= -0.05)$
5: $(\theta_R > 0.08) \wedge (\theta_{T,B} <= 0.42) \wedge (h_R > -0.05)$
6: $(\theta_R > 0.08) \wedge (\theta_{T,B} > 0.42)$ |
| | 4 | 0: $(\dot{\theta}_{T,B} <= -3.83) \wedge (\dot{\theta}_{T,F} <= -0.87)$
1: $(\dot{\theta}_{T,B} <= -3.83) \wedge (\dot{\theta}_{T,F} > -0.87) \wedge (h_R <= -0.08)$
2: $(\dot{\theta}_{T,B} <= -3.83) \wedge (\dot{\theta}_{T,F} > -0.87) \wedge (h_R > -0.08)$
3: $(\dot{\theta}_{T,B} > -3.83) \wedge (\theta_{S,B} <= -0.11) \wedge (h_R <= -0.09)$
4: $(\dot{\theta}_{T,B} > -3.83) \wedge (\theta_{S,B} > -0.11) \wedge (h_R <= -0.09)$
5: $(\dot{\theta}_{T,B} > -3.83) \wedge (\theta_{T,B} <= -0.33) \wedge (h_R > -0.09)$
6: $(\dot{\theta}_{T,B} > -3.83) \wedge (\theta_{T,B} > -0.33) \wedge (h_R > -0.09)$ |
| | 5 | 0: $(\dot{x}_R <= 3.31)$
1: $(\dot{x}_R > 3.31) \wedge (\theta_{S,B} <= 0.01) \wedge (\theta_{T,F} <= 0.57)$
2: $(\dot{x}_R > 3.31) \wedge (\theta_{S,B} > 0.01) \wedge (\theta_{T,F} <= 0.57)$
3: $(\dot{x}_R > 3.31) \wedge (\theta_{T,F} > 0.57)$ |
| | 6 | 0: $(\dot{\theta}_{T,B} <= 1.22) \wedge (\theta_{S,B} <= -0.54)$
1: $(\dot{\theta}_{T,B} > 1.22) \wedge (\theta_{S,B} <= -0.54)$
2: $(\dot{\theta}_{T,B} <= -3.70) \wedge (\theta_{S,B} > -0.54) \wedge (\theta_{T,B} <= 0.16)$
3: $(\dot{\theta}_{T,B} <= -3.70) \wedge (\theta_{S,B} > -0.54) \wedge (\theta_{T,B} > 0.16)$
4: $(\dot{\theta}_{T,B} > -3.70) \wedge \text{(conflict)}$
5: $(\dot{\theta}_{T,B} > -3.70) \wedge (\theta_{S,B} > -0.54)$ |
| | 7 | 0: $(\dot{h}_R <= 0.02) \wedge (\theta_{S,F} <= -0.19) \wedge (\theta_{T,B} <= 0.53)$
1: $(\dot{h}_R <= 0.02) \wedge (\theta_{S,F} <= -0.19) \wedge (\theta_{T,B} > 0.53)$
2: $(\dot{h}_R > 0.02) \wedge (\theta_{S,F} <= -0.19)$
3: $(\dot{\theta}_R <= 0.53) \wedge (\theta_{S,F} <= 0.16) \wedge (\theta_{S,F} > -0.19)$
4: $(\dot{\theta}_R <= 0.53) \wedge (\theta_{S,F} > 0.16)$
5: $(\dot{\theta}_R > 0.53) \wedge (\theta_{S,F} > -0.19)$ |
| | 8 | 0: $(\dot{\theta}_{T,B} <= -0.39) \wedge (\theta_{T,B} <= -0.37)$
1: $(\dot{\theta}_{T,B} <= -0.39) \wedge (\theta_{S,B} <= 0.04) \wedge (\theta_{T,B} > -0.37)$
2: $(\dot{\theta}_{T,B} <= -0.39) \wedge (\theta_{S,B} > 0.04) \wedge (\theta_{T,B} > -0.37)$
3: $(\dot{\theta}_{T,B} > -0.39) \wedge (\theta_{T,B} <= 0.55) \wedge (\theta_{T,F} <= 0.09)$
4: $(\dot{\theta}_{T,B} > -0.39) \wedge (\theta_{T,B} <= 0.55) \wedge (\theta_{T,F} > 0.09)$
5: $(\dot{\theta}_{T,B} > -0.39) \wedge (\theta_{T,B} > 0.55)$ |
| | 9 | 0: $(\theta_{T,F} <= -0.24)$
1: $(\dot{\theta}_{T,F} <= -6.60) \wedge \text{(conflict)}$
2: $(\dot{\theta}_{T,F} > -6.60) \wedge \text{(conflict)}$
3: $(\theta_{T,F} > -0.24) \wedge (h_R <= -0.11)$
4: $(\theta_{T,F} <= 0.32) \wedge (\theta_{T,F} > -0.24) \wedge (h_R > -0.11)$
5: $(\theta_{T,F} > 0.32) \wedge (h_R > -0.11)$ |

Table 30: Logic program of ODE-RNN in locomotion (HalfCheetah).

| Model | Neuron | Logic Program |
|---|---|---|
| CfC | 0 | 0: $(\dot{\theta}_{F,F} <= -9.40)$
1: $(\dot{\theta}_{F,F} > -9.40) \wedge (\dot{\theta}_{T,B} <= -1.68) \wedge (\theta_{F,F} <= 0.23)$
2: $(\dot{\theta}_{F,F} > -9.40) \wedge (\dot{\theta}_{T,B} > -1.68) \wedge (\theta_{F,F} <= 0.23)$
3: $(\dot{\theta}_{F,F} > -9.40) \wedge (\dot{\theta}_{T,F} <= -5.69) \wedge (\theta_{F,F} > 0.23)$
4: $(\dot{\theta}_{F,F} > -9.40) \wedge (\dot{\theta}_{T,F} > -5.69) \wedge (\theta_{F,F} > 0.23)$ |
| | 1 | 0: $(\theta_R <= 0.06) \wedge (\theta_{T,F} <= -0.46)$
1: $(\theta_R > 0.06) \wedge (\theta_{T,F} <= -0.46)$
2: $(\theta_R <= 0.02) \wedge (\theta_{T,F} > -0.46)$
3: $(\theta_R > 0.02) \wedge (\text{conflict})$
4: $(\theta_R > 0.02) \wedge (\theta_{T,F} > -0.46)$ |
| | 2 | 0: $(\dot{\theta}_{T,B} <= 1.92) \wedge (\dot{\theta}_{T,F} <= -0.59) \wedge (\dot{h}_R <= 0.43)$
1: $(\dot{\theta}_{T,B} <= 1.92) \wedge (\dot{\theta}_{T,F} > -0.59) \wedge (\dot{h}_R <= 0.43)$
2: $(\dot{\theta}_{T,B} > 1.92) \wedge (\dot{\theta}_{T,F} <= 5.02) \wedge (\dot{h}_R <= 0.43)$
3: $(\dot{\theta}_{T,B} > 1.92) \wedge (\dot{\theta}_{T,F} > 5.02) \wedge (\dot{h}_R <= 0.43)$
4: $(\dot{\theta}_{S,F} <= 13.53) \wedge (\dot{h}_R > 0.43) \wedge (\theta_{F,B} <= -0.01)$
5: $(\dot{\theta}_{S,F} <= 13.53) \wedge (\dot{h}_R > 0.43) \wedge (\theta_{F,B} > -0.01)$
6: $(\dot{\theta}_{S,F} > 13.53) \wedge (\dot{h}_R > 0.43)$ |
| | 3 | 0: $(\dot{\theta}_{T,F} <= 8.70) \wedge (\theta_{S,F} <= 0.53) \wedge (h_R <= -0.09)$
1: $(\dot{\theta}_{T,F} <= 8.70) \wedge (\theta_{S,F} <= 0.53) \wedge (h_R > -0.09)$
2: $(\dot{\theta}_{T,F} <= 8.70) \wedge (\theta_{S,F} > 0.53)$
3: $(\dot{\theta}_{T,F} > 8.70) \wedge (\theta_R <= 0.20)$
4: $(\dot{\theta}_{T,F} > 8.70) \wedge (\theta_R > 0.20)$ |
| | 4 | 0: $(\theta_{T,F} <= -0.51) \wedge (h_R <= -0.04)$
1: $(\theta_{T,F} <= -0.51) \wedge (h_R > -0.04)$
2: $(\dot{\theta}_{T,B} <= 0.58) \wedge (\theta_{T,F} <= 0.50) \wedge (\theta_{T,F} > -0.51)$
3: $(\dot{\theta}_{T,B} <= 0.58) \wedge (\theta_{T,F} > 0.50)$
4: $(\dot{\theta}_{T,B} > 0.58) \wedge (\dot{h}_R <= -0.44) \wedge (\theta_{T,F} > -0.51)$
5: $(\dot{\theta}_{T,B} > 0.58) \wedge (\dot{h}_R > -0.44) \wedge (\theta_{T,F} > -0.51)$ |
| | 5 | 0: $(\dot{\theta}_{T,B} <= 2.61) \wedge (\theta_{S,F} <= 0.08) \wedge (h_R <= 0.03)$
1: $(\dot{\theta}_{T,B} <= 2.61) \wedge (\theta_{S,F} <= 0.08) \wedge (h_R > 0.03)$
2: $(\dot{\theta}_R <= 0.63) \wedge (\dot{\theta}_{T,B} <= 2.61) \wedge (\theta_{S,F} > 0.08)$
3: $(\dot{\theta}_R > 0.63) \wedge (\dot{\theta}_{T,B} <= 2.61) \wedge (\theta_{S,F} > 0.08)$
4: $(\dot{\theta}_{T,B} > 2.61) \wedge (\theta_{T,F} <= -0.13)$
5: $(\dot{\theta}_{T,B} > 2.61) \wedge (\theta_{T,F} > -0.13)$ |
| | 6 | 0: $(\dot{\theta}_{T,B} <= -0.60) \wedge (\theta_{F,F} <= -0.11)$
1: $(\dot{\theta}_{T,B} > -0.60) \wedge (\theta_{F,F} <= -0.11) \wedge (h_R <= -0.09)$
2: $(\dot{\theta}_{T,B} > -0.60) \wedge (\theta_{F,F} <= -0.11) \wedge (h_R > -0.09)$
3: $(\dot{\theta}_{S,F} <= 11.48) \wedge (\theta_{F,F} > -0.11) \wedge (\theta_{T,B} <= -0.10)$
4: $(\dot{\theta}_{S,F} <= 11.48) \wedge (\theta_{F,F} > -0.11) \wedge (\theta_{T,B} > -0.10)$
5: $(\dot{\theta}_{S,F} > 11.48) \wedge (\theta_{F,F} > -0.11)$ |
| | 7 | 0: $(\dot{\theta}_{T,F} <= -8.38) \wedge (\dot{h}_R <= 0.45) \wedge (\theta_{T,F} <= 0.47)$
1: $(\dot{\theta}_{T,F} > -8.38) \wedge (\dot{h}_R <= 0.45) \wedge (\theta_{T,F} <= 0.47)$
2: $(\dot{h}_R > 0.45) \wedge (\theta_{F,F} <= 0.14) \wedge (\theta_{T,F} <= 0.47)$
3: $(\dot{h}_R > 0.45) \wedge (\theta_{F,F} > 0.14) \wedge (\theta_{T,F} <= 0.47)$
4: $(\theta_{T,F} <= 0.69) \wedge (\theta_{T,F} > 0.47)$
5: $(\theta_{T,F} > 0.69)$ |
| | 8 | 0: $(\dot{\theta}_{T,B} <= -6.34) \wedge (h_R <= -0.02)$
1: $(\dot{\theta}_{T,B} > -6.34) \wedge (h_R <= -0.02)$
2: $(\dot{\theta}_{T,B} > -6.34) \wedge (\text{conflict})$
3: $(\dot{\theta}_{T,F} <= 10.03) \wedge (\theta_{S,F} <= 0.50) \wedge (h_R > -0.02)$
4: $(\dot{\theta}_{T,F} > 10.03) \wedge (\theta_{S,F} <= 0.50) \wedge (h_R > -0.02)$
5: $(\theta_{S,F} > 0.50) \wedge (h_R > -0.02)$ |
| | 9 | 0: $(\dot{\theta}_{T,B} <= 1.14) \wedge (\theta_{S,B} <= -0.06) \wedge (\theta_{S,F} <= 0.46)$
1: $(\dot{\theta}_{T,B} <= 1.14) \wedge (\theta_{S,B} > -0.06) \wedge (\theta_{S,F} <= 0.46)$
2: $(\dot{\theta}_{T,B} <= 1.14) \wedge (\theta_{S,F} > 0.46)$
3: $(\dot{\theta}_{T,B} > 1.14) \wedge (\theta_{T,F} <= 0.56) \wedge (h_R <= -0.10)$
4: $(\dot{\theta}_{T,B} > 1.14) \wedge (\theta_{T,F} <= 0.56) \wedge (h_R > -0.10)$
5: $(\dot{\theta}_{T,B} > 1.14) \wedge (\theta_{T,F} > 0.56)$ |

Table 31: Logic program of CfC in locomotion (HalfCheetah).

| Model | Neuron | Logic Program |
|-------|--------|---------------|
| NCP | 0 | 0: $(\theta_{F,B} <= -0.05)$ 
 1: $(\theta_{F,B} > -0.05)$ |
| | 1 | 0: $(\theta_{F,B} <= -0.04)$ 
 1: $(\theta_{F,B} > -0.04)$ |
| | 2 | 0: $(\theta_{T,B} <= -0.29)$ 
 1: $(\theta_{T,B} > -0.29)$ |
| | 3 | 0: $(\dot{\theta}_{T,F} <= -6.91) \wedge (h_R <= -0.08)$ 
 1: $(\dot{\theta}_{T,F} <= -6.91) \wedge (h_R > -0.08)$ 
 2: $(\dot{\theta}_{T,F} > -6.91) \wedge (\theta_{F,F} <= -0.13)$ 
 3: $(\dot{\theta}_{T,F} > -6.91) \wedge (\theta_{F,F} > -0.13)$ |
| | 4 | 0: $(\dot{\theta}_{F,B} <= -6.37) \wedge (\theta_{T,B} <= -0.36)$ 
 1: $(\dot{\theta}_{F,B} > -6.37) \wedge (\theta_{T,B} <= -0.36)$ 
 2: $(\dot{h}_R <= -0.59) \wedge (\theta_{T,B} <= 0.59) \wedge (\theta_{T,B} > -0.36)$ 
 3: $(\dot{h}_R > -0.59) \wedge (\theta_{T,B} <= 0.59) \wedge (\theta_{T,B} > -0.36)$ 
 4: $(\dot{\theta}_{F,B} <= 0.64) \wedge (\theta_{T,B} > 0.59)$ 
 5: $(\dot{\theta}_{F,B} > 0.64) \wedge (\theta_{T,B} > 0.59)$ |
| | 5 | 0: $(\theta_{F,B} <= -0.02) \wedge (\theta_{T,B} <= -0.40)$ 
 1: $(\theta_{F,B} <= -0.02) \wedge (\theta_{T,B} > -0.40)$ 
 2: $(\theta_{F,B} > -0.02)$ |
| | 6 | 0: $(\dot{\theta}_{F,F} <= -3.61) \wedge (\theta_{F,B} <= -0.06)$ 
 1: $(\dot{\theta}_{F,F} > -3.61) \wedge (\theta_{F,B} <= -0.06)$ 
 2: $(\dot{h}_R <= 0.51) \wedge (\theta_{F,B} > -0.06) \wedge (\theta_{T,F} <= -0.70)$ 
 3: $(\dot{h}_R <= 0.51) \wedge (\theta_{F,B} > -0.06) \wedge (\theta_{T,F} > -0.70)$ 
 4: $(\dot{h}_R > 0.51) \wedge (\theta_{F,B} > -0.06)$ |
| | 7 | 0: $(\theta_{T,B} <= -0.27)$ 
 1: (conflict) 
 2: $(\dot{\theta}_{S,F} <= -8.45) \wedge (\dot{h}_R <= 0.18) \wedge (\theta_{T,B} > -0.27)$ 
 3: $(\dot{\theta}_{S,F} <= -8.45) \wedge (\dot{h}_R > 0.18) \wedge (\theta_{T,B} > -0.27)$ 
 4: $(\dot{\theta}_{S,F} > -8.45) \wedge (\theta_{F,B} <= 0.22) \wedge (\theta_{T,B} > -0.27)$ 
 5: $(\dot{\theta}_{S,F} > -8.45) \wedge (\theta_{F,B} > 0.22) \wedge (\theta_{T,B} > -0.27)$ |
| | 8 | 0: $(\dot{\theta}_{F,B} <= 6.33) \wedge (\dot{\theta}_{T,F} <= 12.79) \wedge (\theta_{S,B} <= -0.04)$ 
 1: $(\dot{\theta}_{F,B} > 6.33) \wedge (\dot{\theta}_{T,F} <= 12.79) \wedge (\theta_{S,B} <= -0.04)$ 
 2: $(\dot{\theta}_{T,F} <= 12.79) \wedge (\theta_{S,B} > -0.04)$ 
 3: $(\dot{\theta}_{T,F} > 12.79)$ |
| | 9 | 0: $(\dot{\theta}_{F,B} <= 6.97) \wedge (h_R <= -0.12)$ 
 1: $(\dot{\theta}_{F,B} <= 6.97) \wedge (\theta_{F,F} <= 0.15) \wedge (h_R > -0.12)$ 
 2: $(\dot{\theta}_{F,B} <= 6.97) \wedge (\theta_{F,F} > 0.15) \wedge (h_R > -0.12)$ 
 3: $(\dot{\theta}_{F,B} > 6.97) \wedge (\dot{\theta}_{S,B} <= -8.53)$ 
 4: $(\dot{\theta}_{F,B} > 6.97) \wedge (\dot{\theta}_{S,B} > -8.53)$ |

Table 32: Logic program of NCP in locomotion (HalfCheetah).

| Model | Neuron | Logic Program |
|-------|--------|---------------|
| FC | 0 | 0: $(\kappa <= 0.00) \wedge (v <= 7.40)$
1: $(\kappa <= 0.00) \wedge (v <= 7.71) \wedge (v > 7.40)$
2: $(\kappa > 0.00) \wedge (v <= 7.71)$
3: $(\kappa <= 0.00) \wedge (d <= 0.12) \wedge (v > 7.71)$
4: $(\kappa <= 0.00) \wedge (d > 0.12) \wedge (v > 7.71)$
5: $(\kappa > 0.00) \wedge (v > 7.71)$ |
| | 1 | 0: $(v <= 7.30)$
1: $(\delta <= 0.00) \wedge (d <= 0.19) \wedge (v > 7.30)$
2: $(\delta <= 0.00) \wedge (d > 0.19) \wedge (v > 7.30)$
3: $(\delta > 0.00) \wedge (d <= 0.29) \wedge (v > 7.30)$
4: $(\delta > 0.00) \wedge (d > 0.29) \wedge (v > 7.30)$ |
| | 2 | 0: $(\delta <= -0.02) \wedge (\kappa <= 0.02) \wedge (\mu <= 0.01)$
1: $(\delta > -0.02) \wedge (\kappa <= 0.02) \wedge (\mu <= 0.01)$
2: $(\kappa > 0.02) \wedge (\mu <= 0.01)$
3: $(\mu > 0.01)$ |
| | 3 | 0: $(\kappa <= -0.00)$
1: $(\kappa > -0.00) \wedge (\mu <= 0.01) \wedge (d <= 0.07)$
2: $(\kappa > -0.00) \wedge (\mu <= 0.01) \wedge (d > 0.07)$
3: $(\kappa > -0.00) \wedge (\mu <= 0.02) \wedge (\mu > 0.01)$
4: $(\kappa > -0.00) \wedge (\mu > 0.02)$ |
| | 4 | 0: $(\kappa <= 0.00) \wedge (\mu <= 0.01) \wedge (v <= 7.66)$
1: $(\kappa <= 0.00) \wedge (\mu <= 0.01) \wedge (v > 7.66)$
2: $(\kappa <= 0.00) \wedge (\mu > 0.01)$
3: $(\kappa > 0.00) \wedge (\mu <= -0.02)$
4: $(\kappa > 0.00) \wedge (\mu <= 0.01) \wedge (\mu > -0.02)$
5: $(\kappa > 0.00) \wedge (\mu > 0.01)$ |
| | 5 | 0: $(\mu <= 0.01) \wedge (v <= 7.49)$
1: $(\delta <= -0.02) \wedge (\mu <= 0.01) \wedge (v > 7.49)$
2: $(\delta > -0.02) \wedge (\mu <= 0.01) \wedge (v > 7.49)$
3: $(\mu <= 0.02) \wedge (\mu > 0.01)$
4: $(\mu > 0.02)$ |
| | 6 | 0: $(\kappa <= -0.01)$
1: $(\delta <= -0.01) \wedge (\kappa > -0.01) \wedge (\mu <= 0.01)$
2: $(\delta > -0.01) \wedge (\kappa > -0.01) \wedge (\mu <= 0.01)$
3: $(\kappa > -0.01) \wedge (\mu > 0.01)$ |
| | 7 | 0: $(\mu <= -0.02)$
1: $(\delta <= 0.02) \wedge (\mu <= 0.01) \wedge (\mu > -0.02)$
2: $(\delta > 0.02) \wedge (\mu <= 0.01) \wedge (\mu > -0.02)$
3: $(\mu > 0.01)$ |

Table 33: Logic program of FC in end-to-end visual servoing (Image-based Driving).

| Model | Neuron | Logic Program |
|---|---|---|
| GRU | 0 | 0: $(d <= -0.06)$
1: $(d <= 0.11) \wedge (d > -0.06)$
2: $(d > 0.11)$ |
| | 1 | 0: $(\mu <= 0.04)$
1: $(\mu <= 0.09) \wedge (\mu > 0.04)$
2: $(\mu > 0.09)$ |
| | 2 | 0: $(\mu <= 0.04)$
1: $(\mu <= 0.10) \wedge (\mu > 0.04)$
2: $(\mu > 0.10)$ |
| | 3 | 0: $(v <= 5.26)$
1: $(\mu <= 0.01) \wedge (v <= 6.88) \wedge (v > 5.26)$
2: $(\mu > 0.01) \wedge (v <= 6.88) \wedge (v > 5.26)$
3: $(v <= 7.34) \wedge (v > 6.88)$
4: $(v > 7.34)$ |
| | 4 | 0: None |
| | 5 | 0: $(v <= 5.26)$
1: $(\mu <= -0.01) \wedge (d <= 0.20) \wedge (v > 5.26)$
2: $(\mu <= -0.01) \wedge (d > 0.20) \wedge (v > 5.26)$
3: $(\mu > -0.01) \wedge (v <= 6.81) \wedge (v > 5.26)$
4: $(\mu > -0.01) \wedge (v > 6.81)$ |
| | 6 | 0: $(\delta <= -0.04)$
1: $(\delta <= 0.05) \wedge (\delta > -0.04) \wedge (v <= 7.81)$
2: $(\delta <= 0.05) \wedge (\delta > -0.04) \wedge (v > 7.81)$
3: $(\delta <= 0.09) \wedge (\delta > 0.05)$
4: $(\delta > 0.09)$ |
| | 7 | 0: $(\kappa <= 0.02) \wedge (\mu <= -0.05) \wedge (d <= 0.61)$
1: $(\kappa <= 0.02) \wedge (\mu > -0.05) \wedge (d <= 0.61)$
2: $(\kappa > 0.02) \wedge (d <= 0.61)$
3: $(\delta <= 0.06) \wedge (\mu <= -0.04) \wedge (d > 0.61)$
4: $(\delta <= 0.06) \wedge (\mu > -0.04) \wedge (d > 0.61)$
5: $(\delta > 0.06) \wedge (d > 0.61)$ |

Table 34: Logic program of GRU in end-to-end visual servoing (Image-based Driving).

| Model | Neuron | Logic Program |
|-------|--------|---------------|
| LSTM | 0 | 0: $(\kappa <= 0.00) \wedge (d <= 0.07)$
1: $(\kappa <= 0.00) \wedge (d > 0.07)$
2: $(\kappa <= 0.00) \wedge (\kappa > 0.00)$
3: $(\kappa > 0.00) \wedge (v <= 7.58)$
4: $(\kappa > 0.00) \wedge (v > 7.58)$ |
| | 1 | 0: $(d <= -0.08) \wedge (v <= 7.54)$
1: $(d <= 0.04) \wedge (d > -0.08) \wedge (v <= 7.54)$
2: $(d <= 0.04) \wedge (v <= 7.68) \wedge (v > 7.54)$
3: $(d <= 0.04) \wedge (v > 7.68)$
4: $(d <= 0.30) \wedge (d > 0.04)$
5: $(d > 0.30)$ |
| | 2 | 0: $(\kappa <= 0.00) \wedge (d <= -0.11)$
1: $(\kappa > 0.00) \wedge (d <= -0.11)$
2: $(d <= 0.03) \wedge (d > -0.11) \wedge (v <= 7.50)$
3: $(d <= 0.03) \wedge (d > -0.11) \wedge (v > 7.50)$
4: $(d <= 0.29) \wedge (d > 0.03)$
5: $(d > 0.29)$ |
| | 3 | 0: $(v <= 7.25)$
1: $(d <= 0.03) \wedge (v <= 7.66) \wedge (v > 7.25)$
2: $(d > 0.03) \wedge (v <= 7.66) \wedge (v > 7.25)$
3: $(\delta <= 0.00) \wedge (v > 7.66)$
4: $(\delta > 0.00) \wedge (v > 7.66)$ |
| | 4 | 0: $(\delta <= 0.05) \wedge (\kappa <= 0.00) \wedge (d <= 0.23)$
1: $(\delta <= 0.05) \wedge (\kappa > 0.00) \wedge (d <= 0.23)$
2: $(\delta <= 0.05) \wedge (d > 0.23)$
3: $(\delta > 0.05)$ |
| | 5 | 0: $(\delta <= 0.04)$
1: $(\delta > 0.04)$ |
| | 6 | 0: None |
| | 7 | 0: $(\delta <= -0.01) \wedge (v <= 7.39)$
1: $(\delta > -0.01) \wedge (v <= 7.39)$
2: $(\kappa <= 0.02) \wedge (d <= 0.01) \wedge (v > 7.39)$
3: $(\kappa > 0.02) \wedge (d <= 0.01) \wedge (v > 7.39)$
4: $(d > 0.01) \wedge (v > 7.39)$ |

Table 35: Logic program of LSTM in end-to-end visual servoing (Image-based Driving).

| Model | Neuron | Logic Program |
|-------|--------|---------------|
| CfC | 0 | 0: $(\delta <= -0.03) \wedge (\mu <= 0.02)$
1: $(\delta > -0.03) \wedge (\mu <= -0.00)$
2: $(\delta > -0.03) \wedge (\mu <= 0.02) \wedge (\mu > -0.00)$
3: $(\mu > 0.02) \wedge (v <= 6.79)$
4: $(\mu > 0.02) \wedge (v > 6.79)$ |
| | 1 | 0: $(\mu <= -0.05)$
1: $(\delta <= -0.03) \wedge (\mu > -0.05)$
2: $(\delta > -0.03) \wedge (\kappa <= 0.00) \wedge (\mu > -0.05)$
3: $(\delta > -0.03) \wedge (\kappa > 0.00) \wedge (\mu > -0.05)$ |
| | 2 | 0: $(\mu <= 0.02) \wedge (d <= 0.12) \wedge (v <= 7.23)$
1: $(\mu <= 0.02) \wedge (d <= 0.12) \wedge (v > 7.23)$
2: $(\mu <= 0.00) \wedge (d > 0.12)$
3: $(\mu <= 0.02) \wedge (\mu > 0.00) \wedge (d > 0.12)$
4: $(\mu > 0.02) \wedge (v <= 6.79)$
5: $(\mu > 0.02) \wedge (v > 6.79)$ |
| | 3 | 0: $(\delta <= -0.04)$
1: (conflict)
2: $(\delta > -0.04) \wedge (d <= 0.16)$
3: $(\delta > -0.04) \wedge (d > 0.16)$ |
| | 4 | 0: $(\kappa <= 0.00) \wedge (\mu <= -0.00)$
1: $(\kappa > 0.00) \wedge (\mu <= -0.00)$
2: $(\mu <= 0.02) \wedge (\mu > -0.00) \wedge (d <= 0.40)$
3: $(\mu <= 0.02) \wedge (\mu > -0.00) \wedge (d > 0.40)$
4: $(\mu > 0.02) \wedge (v <= 6.87)$
5: $(\mu > 0.02) \wedge (v > 6.87)$ |
| | 5 | 0: $(v <= 6.41)$
1: $(\mu <= 0.00) \wedge (v <= 7.15) \wedge (v > 6.41)$
2: $(\mu <= 0.00) \wedge (v > 7.15)$
3: $(\mu > 0.00) \wedge (d <= 0.51) \wedge (v > 6.41)$
4: $(\mu > 0.00) \wedge (d > 0.51) \wedge (v > 6.41)$ |
| | 6 | 0: $(\delta <= -0.00)$
1: (conflict)
2: $(\delta <= 0.04) \wedge (\delta > -0.00) \wedge (v <= 7.18)$
3: $(\delta <= 0.04) \wedge (\delta > -0.00) \wedge (v > 7.18)$
4: $(\delta <= 0.07) \wedge (\delta > 0.04)$
5: $(\delta > 0.07)$ |
| | 7 | 0: $(\delta <= -0.02) \wedge (v <= 7.26)$
1: $(\delta > -0.02) \wedge (v <= 6.56)$
2: $(\delta > -0.02) \wedge (v <= 7.26) \wedge (v > 6.56)$
3: $(\delta <= 0.00) \wedge (v <= 7.55) \wedge (v > 7.26)$
4: $(\delta > 0.00) \wedge (v <= 7.55) \wedge (v > 7.26)$
5: $(v > 7.55)$ |

Table 36: Logic program of ODE-RNN in end-to-end visual servoing (Image-based Driving).

| Model | Neuron | Logic Program |
|---|---|---|
| ODE-RNN | 0 | 0: $(d <= -0.04) \wedge (v <= 5.48)$
1: $(d <= -0.04) \wedge (v <= 7.46) \wedge (v > 5.48)$
2: $(\mu <= 0.02) \wedge (d > -0.04) \wedge (v <= 7.46)$
3: $(\mu > 0.02) \wedge (d > -0.04) \wedge (v <= 7.46)$
4: $(\delta <= 0.05) \wedge (v <= 7.84) \wedge (v > 7.46)$
5: $(\delta <= 0.05) \wedge (v > 7.84)$
6: $(\delta > 0.05) \wedge (v > 7.46)$ |
| | 1 | 0: $(d <= -0.72)$
1: $(d > -0.72) \wedge (v <= 5.80)$
2: $(\delta <= -0.00) \wedge (d > -0.72) \wedge (v > 5.80)$
3: $(\delta > -0.00) \wedge (d > -0.72) \wedge (v > 5.80)$ |
| | 2 | 0: $(v <= 5.00)$
1: $(\delta <= 0.09) \wedge (v <= 6.99) \wedge (v > 5.00)$
2: $(\delta <= 0.09) \wedge (v > 6.99)$
3: $(\delta > 0.09) \wedge (v <= 6.98) \wedge (v > 5.00)$
4: $(\delta > 0.09) \wedge (v > 6.98)$ |
| | 3 | 0: $(d <= -0.04)$
1: $(d <= 0.04) \wedge (d > -0.04)$
2: $(\mu <= -0.11) \wedge (d > 0.04)$
3: $(\mu > -0.11) \wedge (d <= 0.92) \wedge (d > 0.04)$
4: $(\mu > -0.11) \wedge (d > 0.92)$ |
| | 4 | 0: $(\mu <= 0.02) \wedge (d <= 0.60) \wedge (v <= 7.04)$
1: $(\mu <= 0.02) \wedge (d > 0.60) \wedge (v <= 7.04)$
2: $(\mu > 0.02) \wedge (d <= -0.43) \wedge (v <= 7.04)$
3: $(\mu > 0.02) \wedge (d > -0.43) \wedge (v <= 7.04)$
4: $(v <= 7.53) \wedge (v > 7.04)$
5: $(d <= 0.38) \wedge (v > 7.53)$
6: $(d > 0.38) \wedge (v > 7.53)$ |
| | 5 | 0: $(v <= 5.38)$
1: $(\delta <= -0.03) \wedge (d <= 0.10) \wedge (v > 5.38)$
2: $(\delta > -0.03) \wedge (d <= 0.10) \wedge (v > 5.38)$
3: $(\delta <= -0.02) \wedge (d > 0.10) \wedge (v > 5.38)$
4: $(\delta > -0.02) \wedge (d > 0.10) \wedge (v > 5.38)$ |
| | 6 | 0: $(\mu <= 0.01) \wedge (d <= 0.18)$
1: $(\mu <= 0.01) \wedge (d <= 0.47) \wedge (d > 0.18)$
2: $(\delta <= 0.10) \wedge (\mu > 0.01) \wedge (d <= 0.47)$
3: $(\delta > 0.10) \wedge (\mu > 0.01) \wedge (d <= 0.47)$
4: $(d > 0.47) \wedge (v <= 6.60)$
5: $(\mu <= -0.00) \wedge (d > 0.47) \wedge (v > 6.60)$
6: $(\mu > -0.00) \wedge (d > 0.47) \wedge (v > 6.60)$ |
| | 7 | 0: $(v <= 5.15)$
1: $(\mu <= 0.12) \wedge (d <= -0.08) \wedge (v > 5.15)$
2: $(\mu <= 0.12) \wedge (d > -0.08) \wedge (v > 5.15)$
3: $(\mu > 0.12) \wedge (v > 5.15)$ |

Table 37: Logic program of CfC in end-to-end visual servoing (Image-based Driving).

| Model | Neuron | Logic Program |
|-------|--------|---------------|
| NCP | 0 | 0: $(\delta <= -0.05)$
1: $(\delta <= 0.02) \wedge (\delta > -0.05) \wedge (\mu <= -0.01)$
2: $(\delta <= 0.02) \wedge (\delta > -0.05) \wedge (\mu > -0.01)$
3: $(\delta <= 0.09) \wedge (\delta > 0.02) \wedge (\mu <= -0.01)$
4: $(\delta <= 0.09) \wedge (\delta > 0.02) \wedge (\mu > -0.01)$
5: $(\delta > 0.09)$ |
| | 1 | 0: $(\mu <= 0.05)$
1: $(\mu > 0.05)$ |
| | 2 | 0: $(\delta <= 0.02) \wedge (\mu <= -0.03)$
1: $(\delta > 0.02) \wedge (\mu <= -0.03)$
2: $(\delta <= -0.02) \wedge (\mu > -0.03)$
3: $(\delta > -0.02) \wedge (\mu > -0.03)$ |
| | 3 | 0: $(v <= 7.41)$
1: $(v <= 7.72) \wedge (v > 7.41)$
2: $(d <= -0.02) \wedge (v > 7.72)$
3: $(d <= 0.10) \wedge (d > -0.02) \wedge (v > 7.72)$
4: $(d > 0.10) \wedge (v <= 8.05) \wedge (v > 7.72)$
5: $(d > 0.10) \wedge (v > 8.05)$ |
| | 4 | 0: $(v <= 7.45)$
1: $(v <= 7.78) \wedge (v > 7.45)$
2: $(v <= 8.08) \wedge (v > 7.78)$
3: $(v > 8.08)$ |
| | 5 | 0: $(d <= -0.15) \wedge (v <= 7.65)$
1: $(d <= -0.15) \wedge (v > 7.65)$
2: $(\mu <= 0.06) \wedge (d <= 0.04) \wedge (d > -0.15)$
3: $(\mu <= 0.06) \wedge (d > 0.04)$
4: $(\mu > 0.06) \wedge (d > -0.15)$ |
| | 6 | 0: $(\delta <= -0.03) \wedge (v <= 7.73)$
1: $(\delta <= -0.03) \wedge (v > 7.73)$
2: $(\delta > -0.03) \wedge (\mu <= 0.07) \wedge (d <= 0.02)$
3: $(\delta > -0.03) \wedge (\mu <= 0.07) \wedge (d > 0.02)$
4: $(\delta > -0.03) \wedge (\mu > 0.07)$ |
| | 7 | 0: None |

Table 38: Logic program of NCP in end-to-end visual servoing (Image-based Driving).

