# OpenReview forum: "Measuring Interpretability of Neural Policies of Robots with Disentangled Representation"
_robot-learning.org/CoRL/2023/Conference — CoRL 2023 Oral_

### Official Review · Reviewer_tXoG · 2023-07-17

**Confidence:** 2
**Originality:** Fair
**Technical Quality:** Good
**Clarity Of Presentation:** Very Good
**Impact:** 3

**Recommendation:**

Weak Accept: I recommend accepting the paper, but will not argue for my recommendation if the majority of other reviewers have a different opinion.

**Review:**

The paper is clear and to the point at the expense of providing reasoning or motivation for the specific metrics chosen. The limitations outlined are very superficial. The problem space is interesting and applicable to the broader robotics community. Interpretability and guarantees of learned policies are highly sought after.

**Quality Of The Limitations Section:**

Additional details required

**Questions For Rebuttal:**

How do compact networks fit in a world where most planning and robot policies are tending to become larger?
Up to what size models is this method practical?
How does this method fare as complexity of the task increases?
How do interpretability metrics correlate with policy performance? Better performing, but more complex policies may fare worse under interpretability metrics, but better performing at the task.

By design you need a set of states to generate your metrics and decision trees. However the interesting scenarios will usually be those for which you have no examples. How useful is this method in practice given that limitation?

**Robotics Focus:**

Highly relevant to robotics but no hardware experiments

**Summary Of Paper:**

The paper introduces a method to "disentangle" learned policies. Using a set of world states and a trained policy decision trees are extracted as logic programs based on a specific neuron's activation.

These decision trees can then be used to generate metrics defined by the authors in order to measure and compare the interpretability of different models trained for a specific task. The paper focuses on compact neural networks but fails to properly define the scale of what they mean as compact.

**Summary Of Recommendation:**

The paper touches on interpretability of policies for compact networks. Although it is unclear how this may generalize for bigger models / harder tasks, the interpretability and visualization of the decision trees attached to states is quite encouraging.

The metrics presented are interesting and overall the work is highly relevant for the community.

---

> ### Author Response · Authors · 2023-08-15
> **Feedback on the rebuttal**
>
> Dear reviewer tXoG,
>
> Thanks again for your reviews. Based on your reviews, we provided additional experiments and discussion in the rebuttal to address your remaining concerns. Please let us know if there is anything to be further improved in our work. Thank you!!
>
> Sincerely,
>
> Authors

---

### Official Review · Reviewer_75ky · 2023-07-22

**Confidence:** 3
**Originality:** Very Good
**Technical Quality:** Very Good
**Clarity Of Presentation:** Fair
**Impact:** 3

**Recommendation:**

Weak Accept: I recommend accepting the paper, but will not argue for my recommendation if the majority of other reviewers have a different opinion.

**Review:**

I overall found the paper a solid contribution towards interpretability for control policies, and the proposed method demonstrating how to map neural responses and trajectories to decision paths to analyze intuitive. The analysis was also pretty thorough, comparing 3 different metrics across several neural architectures for different control tasks.

I did however find the writing could be improved, particularly more explanation about the high-level purpose of the different metrics, defining notational symbols again as they are split across papers, and more clarity on whether the motivation of the work comes more from a robotics perspective (i.e. its a domain where the authors claim ground truth factor is harder to define) or policy learning perspective (i.e. the authors claim novelty for study neuron-level interpretability of policies). Because of this, it is also unclear how robot-centric this work is, and if CoRL is the right fit. Finally, I feel that a more complete work would run some kind of user study that show why these logical programs are useful, and evaluates whether they actually help human users on some kind of safety checking or decision-making task.

Other comments:

- It seems like the set of works on skill-discovery algorithms could be relevant enough to cite, as they seek to identify sub-trajectories that align with meaningful skills for the task.  Many of them do have a similar API in that they take in trajectories as input, and the key difference is that they just aren't trying to attribute these skills to a particular neuron.
- In the first paragraph of the top of page 2, it's a bit unclear how "Besides, directly observing the neuron response along with sensory information provided as input to the policy can be extremely inefficient and tedious for identifying behaviors and interpreting decision-making" follows from the earlier sentences --> the text here is a bit confusing on what the contrast is.
- Could the authors explain a bit more why compactness is necessary for interpretability? Why is it important that we can attribute behavior to only one particular neuron?
- Lines 124-129 could benefit from more clarity on the process of taking the inverse of a decision tree, as well as maybe re-introducing the notation meanings (z and P) to make it easier to follow .
- It would be great if semantic labels could be attached to the Visuo-Servoing figures for neuron activation to better understand what we should be associating with the neural response in that example, even if they are just the authors' suggestions.

**Quality Of The Limitations Section:**

Additional details required

**Questions For Rebuttal:**


- the authors claim "there is no notion of ground-truth factors in policy learning" --> is this a property of policy learning, or the domain of robotics/control? It seems like the issue is more the latter, as surely there are simply domains where there would be a clear sense of ground truth factors for a policy?


**Robotics Focus:**

Highly relevant to robotics but no hardware experiments

**Summary Of Paper:**

The authors propose a novel way of studying interpretability of learned policies for robot learning from the perspective of disentangled representations. They aim to learn a logical program for each neuron by training a decision tree over trajectories containing state information and individual neural response. At inference time, the decision tree takes as input a neural response, and generates decision paths that correspond to logical programs to aid interpretability. The authors then suggest 3 metrics to evaluate these decision paths (variance of neural response, mutual information with ground truth factors, and modularity). They then compare different neural architectures on Pendulum, HalfCheetah, and Visual Servoing tasks .

**Summary Of Recommendation:**

Overall, I think the paper is solid and added clarifications to my points above would help increase my score.

---

> ### Author Response · Authors · 2023-08-15
> **Feedback on the rebuttal**
>
> Dear reviewer 75ky,
>
> Thanks again for your reviews. Based on your suggestions, we provided additional experiments and clarification in the rebuttal. Please let us know if these address your remaining concerns or if there is anything to be further improved in our work. Thank you!!
>
> Sincerely,
>
> Authors

---

### Official Review · Reviewer_Z6qk · 2023-07-23

**Confidence:** 2
**Originality:** Very Good
**Technical Quality:** Good
**Clarity Of Presentation:** Good
**Impact:** 4

**Recommendation:**

Strong Accept: I recommend accepting the paper and will argue for my recommendation even if other reviewers hold a different opinion.

**Review:**

Strengths:
- The main text and appendices are highly detailed.
- The findings in the paper were interesting to me/sparked a lot of ideas. I imagine many readers will feel the same way. (e.g., the behavior of command neurons in locomotion experiments)
- The idea of modeling neurons with decision trees is elegant and can be repurposed across many kinds of robotic tasks.

Weaknesses:
- The paper is motivated by interpretability but it's not clear to me that the metrics evaluate interpretability with respect to human judgments. Although three of the metrics are derived from an evaluation of human interpretability, they are ultimately re-interpreted for the paper. Adding a small interpretability experiment with human subjects would make for a much stronger claim.
- The presentation of the metrics could be made more clear. In particular the intuition behind/motivation for some of the metrics could be made more clear. This is especially true for MIG.

Nits (not material to review, just recommendations for revisions):
- Add a caption to the first figure
- Slight grammatical errors in L35, L54
- Defining K_i as the decision paths more explicitly somewhere would help with readability
- The first cited work for MIG (numbered [8] in the bibliography) does not discuss mutual information. As a reader I was confused how this paper was inspirational for this section.

**Quality Of The Limitations Section:**

Limitations are addressed clearly

**Questions For Rebuttal:**

Please address the weaknesses listed above. Additionally:
- L143: What does it mean that the set contains time steps that correspond to the behavior of P^i_k? Specificall, how is "corresponding behavior determined?

**Robotics Focus:**

Relevant but unlikely to deploy to hardware in near future

**Summary Of Paper:**

The goal of this paper is to develop methods for interpreting learned policies. To that end, the paper proposes learning a decision tree to model each neuron of a policy network. The interpretability is evaluate with three metric that they introduce and three metrics proposed in prior work.

The three metrics that they introduce are:
1. The average **variance** of a finite set of decision paths across neurons
2. The **mutual information gap** to compute the disentanglement of different decision paths among neurons
3. The **modularity** of each neuron with respect to different decision paths

The authors measure the interpretability of 6 different architectures on classical control, locomotion, and autonomous driving tasks.

**Summary Of Recommendation:**

I am recommending the paper for acceptance because the authors explore an elegant idea thoroughly. Although the robotics focus is not direct (as the experiments are in sim) I think many of the members of the corl community will find the method relevant and the findings interesting.

---

> ### Author Response · Authors · 2023-08-15
> **Feedback on the rebuttal**
>
> Dear reviewer Z6qk,
>
> Thanks again for your reviews and recognition on the quality and impact of our work. Based on your comments, we provided additional experiments and discussion in the rebuttal to further strengthen the paper. Please let us know if there is anything to be further improved in our work. Thank you!!
>
> Sincerely,
>
> Authors

---

### Official Review · Reviewer_BCVj · 2023-07-31

**Confidence:** 4
**Originality:** Good
**Technical Quality:** Excellent
**Clarity Of Presentation:** Good
**Impact:** 3

**Recommendation:**

Weak Accept: I recommend accepting the paper, but will not argue for my recommendation if the majority of other reviewers have a different opinion.

**Review:**

Quality: Overall, the paper is well-written, equations appear correct, and the diagrams are aesthetically pleasing and provide a vision for how this method could be used in a real-world robotics application.

Clarity: The paper’s explanations of the motivation for the method and the theory used are mostly clear, and most concerns are addressed by the rebuttal. The remaining concerns I have are about the amount of cognitive overload it takes to qualitatively interpret the decision tree branch analysis. I still think a video of the trajectories would make this a bit clearer, but in the meantime, the diagrams the authors produced are of good quality.

Originality: The paper claims to introduce a novel set of quantitative metrics for interpretability; however, all of the metrics appear to be taken from (cited) established works on disentanglement, decision trees, or interpretability. The paper’s most original contribution appears to be the use of extracted logical programs as a factor of variation for disentanglement in simulated control settings.

Significance: Interpretability methods like the one presented in this paper are important and relatively underexplored in robotics, where it could be highly impactful for debugging real-world issues and monitoring safety-critical applications. For future work the authors will need to address the limitations with using privileged states used in the logic programs which are not available to the policy.

**Quality Of The Limitations Section:**

Additional details required

**Questions For Rebuttal:**

Can you clarify the exact reason why we need the inverse classifier?
What are the specific details of the classifier training process (technique, optimizer, classification criterion, dataset size)?
What are the specific details of the decision tree training (similar details to above)?
Can you clarify what are the pre-determined grounding states of the program and how these are distinct from the states observed in the environment?
Is the same data used for policy learning used in decision tree/classifier training?
What are the error bars on the numbers cited in the results?
Do the “failure states” for the locomotion task discovered by the decision tree actually correlate to episode terminations?
Could you use other sources of ground truth as factors of variation, for example, a handcrafted PID controller for the pendulum task?

**Robotics Focus:**

Highly relevant to robotics but no hardware experiments

**Summary Of Paper:**

This work presents a disentanglement method for compact NNs in policy learning. There is no ground truth for the factors of variation, since emergent behaviors are unknown a priori, so decision trees from observable states to specific neural responses are extracted from the policy, along with an inverse mapping predicting paths through the decision tree from neural responses. The metrics for quantifying interpretability are: Neuron-response variance, the concentration of the neuron response for a particular decision path. Mutual Information Gap, modified by an additional calibration term, the gap between the highest and second highest mutual dependencies. Modularity, 1 - the mutual information between a neuron response and all non-matched decision paths. Inverse correlation between metrics and decision tree quality metrics. Empirical results and qualitative illustrations of the interpretability of this method are shown for pendulum, locomotion, and visual servoing environments.

**Summary Of Recommendation:**

The paper's premise is interesting and potentially useful to robotics. The authors have integrated together an impressive body of knowledge of decision trees, logical programs, neural networks, reinforcement learning, and interpretability metrics. They have clearly conducted extensive experiments using this technique, which shows promise for future work. Future work will need to show how it can be extended to different neural network architectures, how to deal with privileged states, and clearer presentations of how the branch analysis and discovered factors of variation relate to intuitively critical states in the environment trajectory.

---

### Author Response · Authors · 2023-08-13
**Global Response**

We thank all reviewers for their thoughtful and constructive feedback. We are encouraged to hear the reviewers’ positive feedbacks including,
- the paper is well-written and mostly clear (BCVj), highly detailed (Z6qk), clear and to the point (tXoG).
- the problem space is interesting and applicable to the broader robotics community (tXoG) and could be highly impactful for debugging real-world issues and monitoring safety-critical applications, which will be found relevant by the CoRL community (BCVj).
- the proposed method is interesting (BCVj), elegant (Z6qk), and highly sought after (tXoG) with solid contribution towards interpretability for control policies (75ky).
- the results, findings and analysis provide a vision for how this method could be used in a real-world robotics application (BCVj), are interesting to spark a lot of ideas (Z6qk) and thorough (75ky).

In response to feedback, we provide individual responses below to address the remaining concerns from each reviewer and a revised manuscript with changes highlighted in red to improve clarity or provide further discussion. Briefly, we summarize the additional experiments and revision to the paper,
- A simple user study to verify the effectiveness of the proposed metrics and extracted logic programs.
- Included error bars for all experiments with multiple random seeds.
- Quantitative analysis on the failure modes in locomotion.
- An experiment on a transformer-based policy to demonstrate the potential extension to larger-scale models.
- Revision on the original papers on high-level intuition, improved exposition, and minor errors.

For more details, please check individual responses. We thank all reviewers’ for their time and efforts! We hope our responses have persuasively addressed all remaining concerns. Please don’t hesitate to let us know of any additional comments or feedback on improvement.

Thanks, authors

---

### Author Response · Authors · 2023-08-14
**Thank you and we are looking forward to your feedback on the rebuttal!**

Dear AC and all reviewers:

Thanks again for all the insightful comments and feedbacks, which helped us improve the paper's quality and clarity.

We went through the reviews carefully and made dedicated efforts to address all concerns in the rebuttal.

We would love to convince you of the merits of the paper. Please do not hesitate to let us know if there is any additional experiment or clarification that we can offer to make the paper better. We appreciate your comments and advice.

Best,

Author

---

### Decision · Program_Chairs · 2023-08-30

**Decision:**

Accept (Oral)

**Comment:**

This paper introduces an interesting approach for measuring interpretability of robot policies. Overall reviewers found the paper to be well-written and a clear contribution to the field. As such, I am happy to recommend the paper be accepted.